# *In vitro* and *in silico* investigation of effects of antimicrobial peptides from Solanaceae plants against rice sheath blight pathogen *Rhizoctinia solani*

Tijjani Mustapha[1], Shefin B[2☯], Talha Zubair[3☯], Rajesh B. Patil[4], Bhoomendra A. Bhongade[5], Jaiprakash N. Sangshetti[6], Aniket Mali[7], Balogun Joshua Babalola[1], Abu Tayab Moin[8], Tofazzal Islam[9]*

1 Department of Biological Sciences, Federal University, Dutse, Nigeria, 2 Jawaharlal Nehru Tropical Botanic Garden and Research Institute, Trivandrum, India, 3 Notre Dame College, Dhaka, Bangladesh, 4 Department of Pharmaceutical Chemistry, Sinhgad Technical Education Society's, Sinhgad College of Pharmacy, Vadgaon (Bk), Pune, Maharashtra, India, 5 Department of Pharmaceutical Chemistry, RAK College of Pharmacy, RAK Medical & Health Sciences University, Ras Al Khaimah, United Arab Emirates, 6 Y. B. Chavan College of Pharmacy, Dr. Rafiq Zakaria Campus, Rauza Baugh, Aurangabad, (MS), India, 7 Cancer Research Lab, Interactive Research School for Health Affairs (IRSHA), Bharati Vidyapeeth (Deemed to be University), Pune, Maharashtra, India, 8 Faculty of Biological Sciences, Department of Genetic Engineering and Biotechnology, University of Chittagong, Chattogram, Bangladesh, 9 Institute of Biotechnology and Genetic Engineering (IBGE), Bangabandhu Sheikh Mujibur Rahman Agricultural University (BSMRAU), Gazipur, Bangladesh

☯ These authors contributed equally to this work.
* tofazzalislam@bsmrau.edu.bd

**Data Availability Statement:** The data associated with this paper are included as Table and Figures.

## Abstract

*Rhizoctonia solani*, the causative agent of sheath blight disease in rice, poses a significant threat to agricultural productivity. Traditional management approaches involving chemical fungicides have been effective but come with detrimental consequences for the ecosystem. This study aimed to investigate sustainable alternatives in the form of antifungal peptides derived from Solanaceous plant species as potential agents against *R. solani*. Peptide extracts were obtained using an optimized antimicrobial peptide (AMP) extraction method and desalted using the solid-phase extraction technique. The antifungal potential of peptide-rich extracts from *Solanum tuberosum* and *Capsicum annum* was assessed through in vitro tests employing the agar well diffusion method. Furthermore, peptide-protein docking analysis was performed on HPEPDOCK and HDOCK server; and molecular dynamics simulations (MDS) of 100 ns period were performed using the Gromacs 2020.4. The results demonstrated significant inhibition zones for both extracts at concentrations of 100 mg/mL. Additionally, the extracts of *Solanum tuberosum* and *Capsicum annum* had minimum inhibitory concentrations of 50 mg/mL and 25 mg/mL, respectively with minimum fungicidal concentrations of 25 mg/mL. Insights into the potential mechanisms of key peptides inhibiting *R. solani* targets were gleaned from *in-silico* studies. Notably, certain AMPs exhibited favorable free energy of binding against pathogenicity-related targets, including histone demethylase, sortin nexin, and squalene synthase, in protein-peptide docking simulations. Extended molecular dynamics simulations lasting 100 ns and MM-PBSA calculations were performed

**Funding:** The author(s) received no specific funding for this work.

**Competing interests:** The authors have declared that no competing interests exist.

on select protein-peptide complexes. AMP10 displayed the most favorable binding free energy against all target proteins, with AMP3, AMP12b, AMP6, and AMP15 also exhibiting promising results against specific targets of *R. solani*. These findings underscore the potential of peptide extracts from *S. tuberosum* and *C. annum* as effective antifungal agents against rice sheath blight caused by *R. solani*.

## Introduction

Sheath blight disease of rice (*Oryza sativa* L.) caused by *R. solani* Kuhn remains one of the significant biotic constraints limiting crop production in rice-producing areas worldwide [1]. To date, no rice variety with a sheath blight-resistant trait has been discovered, but only varieties with polygenic resistance that become susceptible after a few years are available [2–4]. AG1-IA is the most economically important plant disease-causing organism among *R. solani* anastomosis groups (AG) [5]. Under favorable environmental conditions for sheath blight disease, yield losses exceeding 50% may occur. For instance, in key rice-producing regions of Asia, sheath blight caused by R. solani affected over 20 million hectares, resulting in a grain loss exceeding 6 million tons of rice [5,6].

Plant antimicrobial peptides (AMPs) are naturally occurring metabolites characterized by low to negligible levels of toxicity, positive charge, and the presence of disulfide bonds. Their unique conformational structure enables them to withstand diverse environmental conditions [7,8]. However, it has been noted that chemical antimicrobial agents such as synthetic fungicides commonly control phytopathogens. The environmental hazards caused by such chemicals and the evolution of fungicide-resistant strains of plant pathogenic fungi led to global advocacy for using alternatives to plant disease control with sustainable and eco-friendly attributes. However, plants such as members of the family of Solanaceae are promising sources of natural metabolites such as AMPs with a wide range of antimicrobial activities [8].

Solanaceae, commonly called the nightshade or potato family, is an *Angiospermeae* family in the plant kingdom consisting of 98 genera encompassing about 2,715 identified species. They are cosmopolitan in distribution with habitat range of both temperate and tropical climates (Singh, 2010). Among the genera of the *Solanaceae* family, *Capsicum* and *Solanum* genera accommodate species with antimicrobial peptides present in great quantity [8]. Peptides such as those that resemble thionine and vicilin, lectins, protease inhibitors, defensins, and snaking peptides are responsible for the antimicrobial activities, which are found to be associated with Solanaceous species in significant concentrations [8]. Peptide extracts from several species of *Capsicum* inhibit the growth, enzyme and pathogen-derived protein activities of *Colletotrichum* species, *Alternaria solani*, and *Fusarium* species, to mention a few [9–11]. While evidence substantiates the antimicrobial activity of antimicrobial peptides (AMPs) derived from Solanaceae plants against various pathogens, including fungi, the exact mechanisms underlying their efficacy against *R. solani* remain incompletely elucidated.

During infection, fungal pathogens commonly express proteins responsible for pathogenicity, virulence, and other metabolic pathways that influence their attack. Prabhukarthikeyan *et al.* (2022) [4] used a quantitative liquid chromatography and mass spectrometry (LC-MS/MS)-based proteomic approach to identify several proteins in *R. solani* that promote pathogenicity and virulence, regulate vegetative growth, morphogenesis, sexual development, sporulation, and regulate stress conditions and crucial molecular processes within the pathogen's biological systems. The study found that Histone Demethylation Protein-1, Heat Shock Protein-70 (HSP70), Sortin Nexin-4, and Squalene synthase proteins are associated with the

functions mentioned above (Prabhukarthikeyan *et al.*, 2022 [4]). Therefore, exploring the molecular mechanisms by which these AMPs exert their antifungal effects on *R. solani* could represent a significant gap in current knowledge. This could involve investigating the binding interactions between AMPs and specific molecular targets within the fungus, as well as elucidating the impact of these interactions on key cellular processes related to fungal growth, development, and pathogenicity. It can be hypothesized that peptide extracts derived from *C. annum* and *S. tuberosum* exhibit antimicrobial properties against *R. solani*. Furthermore, these peptide extracts could interact with pivotal proteins implicated in the pathogenicity and virulence of *R. solani*, namely Histone Demethylase, Sortin Nexin-4, and Squalene synthase, potentially impeding their function.

This study aimed to isolate the pure culture of *R. solani* and investigate the antimicrobial effects of peptide extracts from *C. annum* and *S. tuberosum* by examining their *in vitro* antifungal effects on *R. solani*. Furthermore, to enhance comprehension of the selected AMPs' potential impact on the pathogenicity and virulence-associated proteins of *R. solani*, namely Histone Demethylase, Sortin Nexin-4, and Squalene synthase from these plants, molecular docking, molecular dynamics simulation studies, and MM-PBSA calculations were conducted.

## Materials and methods

### Source of inoculum

A field survey was conducted in some rice fields around the Dutse area of Jigawa State, Nigeria, located at the geographical coordinate of 11˚43'59.99" N 9˚17'15.00" E, and rice plants showing sheath blight disease were identified. Symptoms of the disease mentioned in Park *et al.*, 2008 [3]; and Moni *et al.*, 2016 [12], were used as an identification guide for collecting *R. solani* infected plants. The collected diseased plant materials were sealed and labelled in paper envelopes and then transported to the laboratory for processing.

### Sterilization of materials and media preparation

The collected diseased plant material was subjected to surface sterilization to remove contaminants. A 1% sodium hypochlorite solution and sterile distilled water were used to surface sterilize and rinse the sample before incubation. The equipment to be used, as well as the working environment, was sterilized using 95% ethyl alcohol to ensure an aseptic condition. Potato dextrose agar (PDA) was prepared according to the manufacturer's instruction, autoclaved at 121˚ C for 15 minutes, and a few drops of acetic acid were added to avoid bacterial growth and to enhance the growth of *R. solani* [13].

### Isolation and purification of *R. solani*

The pathogen was isolated through the direct plate method described by Park *et al.* (2008) [3] and Senanayake *et al.* (2020) [13]. About 5 mm of the infected material was sliced through the free-hand sectioning technique and placed on a solidified PDA plate. The inoculated plate was incubated at 20–25˚C and observed for fungal growth. The pathogen was sub-cultured and purified according to the standard protocol of the hyphal tip culture technique described in [12] and cultivated on PDA to obtain the pure isolate of *R. solani*. Plates were repeatedly sub-cultured until uniform pure isolate was obtained.

### Identification of pathogen

Slide culture techniques were employed to grow the pure isolate on a 5 mm square block of PDA on a microscope slide, then covered with a cover slip and incubated at room temperature

for 48 hours. The cover slip from the incubated slide was placed on a clean and grease-free, containing a few drops of lactophenol cotton blue stain, mounted on the microscope, and observed the morphology at low power (X10) and high-power (X40) magnifications. Microscopic identification involved observing the features described by [12,14]. The macroscopic identification was made from the observation of colony morphology, mycelial color, superficial sclerotia, and texture, as described in [12].

## Standardization of inoculum

About 200 mL of mycelial suspension of the pure isolate of *R. solani* was prepared in potato dextrose broth (PDB) and homogenized according to the protocol adopted by Park *et al*. (2008) [3]. The homogenized mycelial suspension for the antifungal test was gently added to 2-3mL 0.9% sterile normal saline water, and the turbidity was adjusted until $10^5$ CFU/mL obtained using a spectrophotometer at 0.08 to 0.12 at 625 nm absorbance [15].

## Plant material collection and sterilization

The chili (*Capsicum annuum*) and Potato (*Solanum tuberosum*) sample was collected from agricultural fields around Dutse district, Jigawa State, Nigeria. The fresh samples were identified through comparison with the reference samples of *C. annum* and *S. tuberosum* deposited in the Herbarium section of the Department of Biological Sciences, Federal University Dutse. Samples were surface sterilized in 1% sodium hypochlorite solution, then rinsed several times using distilled water and dried at room temperature until constant weight was recorded.

## Sample homogenization and extraction

The dried plant materials from the two plant species were pulverized into powder using laboratory mortar and pestle and then sieved. The optimized method of AMP extraction was used as an extraction procedure in this study [16–18]. Thus, 50 g from the sample was mixed with 500 mL of 10% methanolic acid and left for 1h at room temperature with concurrent shaking. The mixture was separated through sieving and centrifuged at 4700 rpm for 10 minutes. The supernatant recovered was concentrated under reduced pressure using a rotary evaporator. Sample precipitation was done by reconstitution of 50 mL of the concentrated supernatant in 7.1 mL of cold acetone (-70˚ C) and kept at 4˚ C for 6-8h, then precipitated fraction was obtained by centrifugation at 4700 rpm for 10min. The precipitate was desalted by solid phase extraction, evaporated and freeze-dried to obtain the peptide-rich extract used for antifungal bioassay [17].

## Preparation of different concentrations

Different concentrations of the peptide extracts from *C. annum* and *S. tuberosum* were prepared using the standard protocol of the double-fold dilution method. Thus, 2 mL of 100 mg/mL, 50 mg/L and 25 mg/mL concentrations of both extracts were prepared using DMSO as a diluent and stored at 4˚ C before use.

## *In vitro* antifungal bioassay

**Determination of zone of inhibition.** The protocol of the well diffusion method, according to Balouiri, Sadiki and Ibnsouda, 2016 was used in this study to determine the inhibition zone by the different concentrations of the extracts. A prepared PDA plate was allowed to solidify and inoculated gently with the standard mycelial suspension using a sterile swab stick and settle for a few minutes. A 6 mm diameter hole was made on the inoculated PDA plates,

filled with a few drops of the extracts, and allowed the extracts to diffuse. The plates were incubated at 20–24° C for 2–3 days, after which the diameter zone of inhibition was recorded using the transparent meter rule and expressed in millimeters. DMSO was used as a negative control throughout the bioassay, while Azoxystrobin was used as a positive control.

**Determination of minimum inhibitory concentration (MIC) and minimum fungicidal concentration (MFC).** The MIC was obtained based on Balouiri, Sadiki and Ibnsouda, 2016 through the broth macrodilution method [19] with major modifications. Briefly, 0.5 mL of PDB was poured into a sterile different test tube containing 0.5 mL from each of the extracts respectively. This was mixed with 1 mL of the standard inoculum, blocked the test tubes with cotton wool, and incubated at 20–24°C for 3 days. The MIC values were recorded based on the visual observation of the turbidity formed due to mycelial growth. The MFC was determined by subculturing the broth from each test tube on fully labelled PDA plates and incubated. The plates showing no fungal growth were determined to be the MFC value [15]. All experiments were carried out in triplicate under a completely randomized design (CRD).

## Protein-peptide docking analysis

The sequences of target proteins were retrieved from the Uniprot database (www.uniprot.org). Specifically, the *R. solani* specific sequence of Histone demethylase (Accession number: L8X0Z2), sortin (Accession number: A0A0K6FTR5), and squalene (Accession number: L8WRK0) were retrieved and used to construct the homology models using modeler 9.25 program [20] using multiple template structures. The resulting homology models with the lowest DOPE scores were further used in protein-peptide docking. The structures of AMP from the Solanaceae family were predicted from the PEP-FOLD3 server [21,22]. The blind docking of the peptide structures with the target proteins was performed on the HPEPDOCK server [23]. However, in the case of peptides having sequence lengths longer than 20, the HDOCK server [24] was employed.

## Molecular dynamics studies and MM-PBSA calculations

From the docking studies, the five best protein-peptide complexes for each protein with the best docking scores were selected for molecular dynamics simulations (MDS). The MD simulations of 100 ns period were performed using the Gromacs 2020.4 [25] package on the HPC cluster at Bioinformatics Resources and Applications Facility (BRAF), C-DAC, Pune. The input topologies of protein and AMP were prepared with CHARMM-36 force field parameters [26,27]. During MDS system preparation, each complex was placed in a box of dodecahedron unit cells, maintaining the 1 nm distance between the system and the edges of the box. Subsequently, the systems were solvated with the TIP3P water model [28] and neutralized with suitable counter-ions, such as sodium or chloride ions. The energy minimization with the steepest descent algorithm employing the force constant threshold of 1000 KJ $mol^{-1}$ $nm^{-1}$ was performed on the resulting systems. The constant volume and constant temperature conditions (NVT) equilibration, where the temperature of 300 K was achieved with a modified Berendsen thermostat [29], was performed with 1 ns simulation. Subsequent 1 ns equilibration was performed at constant volume and constant pressure (NPT) conditions, where the pressure of 1 atm was achieved with Berendsen barostat [30]. The production phase MD simulations of 100 ns on each equilibrated system were performed, where the temperature was maintained with a modified Berendensen thermostat and pressure was maintained with Parrinello-Rahman barostat [31]. The restrain on covalent bonds was achieved with the LINCS algorithm [32] and during simulation the long range electrostatic energies were computed with Particle Mesh Ewald (PME) method [33] with the cut-off of 1.2 nm.

In the production phase of MDS, the trajectories stored at each 10 ps were treated to remove the periodic boundary conditions (PBC). The MDS analysis of deviations in the C-$\alpha$ atom of protein and AMP from the initial positions was investigated in terms of root mean square deviations (RMSD). The fluctuations in the side chain atom of protein and AMP were analysed regarding root mean square fluctuation (RMSF). The compactness of each protein-peptide complex was analyzed by estimating the radius of gyration (Rg). The hydrogen bond analyses where the hydrogen bonds formed between the protein and AMP were performed, and trajectories at different time intervals were manually inspected for key hydrogen bonds formed. The mean smallest distance between residue pairs was analyzed through the gmx_mdmat program, and the residue-wise contact maps [34] were constructed. The principal component analysis (PCA) [35] was performed to study the path of motions in protein and AMP, where the covariance matrix for the C-$\alpha$ atom of protein and AMP was generated, which after diagonalizing gave the eigenvectors and eigenvalues. The path of motion was interpreted from the eigenvectors while the mean square fluctuations were interpreted from the eigenvalues. The first two principal components (PC1 and PC2) were used to derive the lowest energy state conformations of protein-peptide complexes and to perform Gibb's free energy landscape (Gibb's FEL) analysis [36]. The secondary structural changes were captured through the definition of protein secondary structure (DSSP) program [37,38]. The analysis of fluctuations and displacements of side chains of protein-peptide complexes was assessed through the dynamical cross-correlation matrix (DCCM) [39] analysis.

The trajectories extracted from the reasonably stable MD simulation period 75 ns to 100 ns at each 500 ps were subjected to Poisson Boltzmann surface area continuum solvation (MM-PBSA) calculations [40,41].

The protein structures were rendered in ChimeraX [42], PyMOL(Schrödinger, LLC, 2015) [43], and VMD [44], and graphs were plotted in the XMGRACE (Turner, 2005) interface [45]. Gibb's FEL plots were generated using a Python-based Matplotlib package [46]. The DCCM analysis was performed in the R statistical program [47] with the Bio3D package [48].

### Statistical analysis

The data gathered from the assessment of *in vitro* effects of the peptide extracts was subjected to analysis using One Way Analysis of Variance (ANOVA) via the IBM-SPSS statistical package version 20, with a significance level set at P$\leq$0.05. *Post hoc* analysis was conducted utilizing Tukey's honest significance test.

### Ethical approval

No human or animal subjects were used in this plant-microbe study. Therefore, no ethical approval was required in the laboratory experiments and computational analysis.

### Results

Table 1 shows that *C. annum* peptide extract at 100 mg/mL had the greatest effect, with an inhibition zones of 40.67 mm (**Fig 1**). A relatively lower inhibition of *R. solani* fungus was obtained at 50 and 25 mg/mL concentrations, with 38.67 and 32.67 mm zones of inhibition. Mycelial growth inhibition on *R solani* was found to be significant between concentrations tested, with no significant difference between 50 and 100 mg/mL but significant between 25 mg/mL and other concentrations used. In *S. tuberosum* peptide extract from tubers, the highest mean zone of inhibition was 46.00 mm at 100 mg/mL, followed by 40.67 mm at 50 mg/mL and 36.33 mm at 25 mg/mL. The mean zone of inhibition difference between 50 and 100 mg/mL was not statistically significant, whereas 25 mg/mL was significantly different from all concentrations except the positive control.

**Table 1. Effects (zone of inhibition, mm) of peptide rich extracts from *C. annum* and *S. tuberosum* against *R. solani*.**

| Extracts | Concentrations mg/mL | | | | Positive control (Azoxystrobin) |
|---|---|---|---|---|---|
| | 0 | 25 | 50 | 100 | |
| *C. annum* | 0.00[a] | 32.67[b] | 38.67[c] | 40.67[c, d] | 38.67[c, e] |
| *S. tuberosum* | 0.00[a] | 36.33[b] | 40.67[c] | 46.00[c, d] | 38.67[b, e] |
| P-value≤ 0.05 | | | | | |

Table 2 shows that *R. solani* mycelial growth was inhibited at 50 and 100 mg/mL concentrations in *C. annum* extract. Similarly, in all concentrations tested, *S. tuberosum* extract completely inhibited mycelial growth. Growth inhibition was not inhibited by the negative control (DMSO), but the positive control inhibited it.

All treatments except 25 mg/mL and the negative control had a fungicidal effect on *R. solani* treated with the extract (Table 3). Mycelial growth was observed in *S. tuberosum* at all concentrations except the negative control. *C. annum* and *S. tuberosum* peptide extracts had MFCs of 50 and 25 mg/mL, respectively.

## Protein-peptide docking

A total of 17 peptides were docked (global docking) against three modelled proteins. The best docking scores for the best-ranked models of HPEPDOCK and HDOCK results are presented in Table 4.

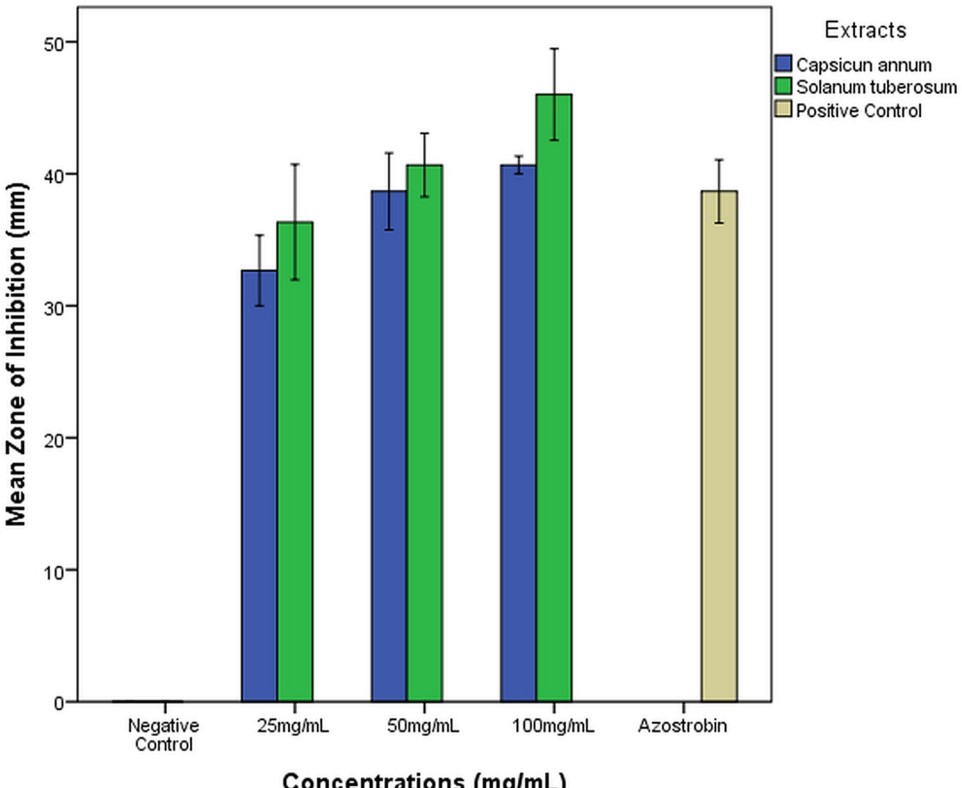

**Fig 1. Mean zone of Inhibition of Different concentrations of *C. annum* and *S. tuberosum* of R. solani.** The error bars represent standard errors of the mean.

**Table 2. Effect of peptide rich extracts from *C. annum* and *S. tuberosum* on the Minimum Inhibitory Concentration (MIC) on *R. solani*.**

| Extracts | Concentrations mg/mL | | | | Positive control (Azoxystrobin) |
|---|---|---|---|---|---|
| | 0 | 25 | 50 | 100 | |
| *C. annum* | ─ | ─ | ++ | + | + |
| *S. tuberosum* | ─ | ++ | + | + | + |

Note: (─) = Turbidity observed (MIC Negative); (+) = No turbidity observed (MIC Positive) and (++) = MIC Value.

## Molecular dynamics simulations and MM-PBSA calculations

The top five complexes for each protein were subjected to molecular dynamics simulations and MM-PBSA calculations.

## Root mean square deviations (RMSD) analysis

The results of RMSD in the C-α atom of respective proteins when bound to the peptides and the C-α atom of peptide chains bound to the respective proteins are shown in **Fig 2**.

The RMSD in C-α atom of histone demethylase in bare protein and AMP bound complexes stabilize after around 50 ns simulation period. The RMSD in AMP-10 bound histone demethylase is lowest compared to bare protein and other complexes with an average RMSD of 0.8 nm (**Fig 2**). The AMP-15 bound histone demethylase showed higher RMSD than bare protein and other complexes with an average RMSD of 1.5 nm. However, the RMSD in the C-α atom of AMP-15 is the lowest among the complexes, with an average of around 0.2 nm. The RMSD in the C-α atom of AMP-10 stabilizes after around 60 ns with an average of around 0.4 nm, and the AMP-1 has the highest RMSD amongst all the complexes with an average of around 0.41 nm. The RMSD in the C-α atom of sortin within a narrow range of 0.5 nm to 1.0 nm, where the sortin bound to AMP-12a has the lowest RMSD and AMP-12b bound has the highest RMSD with averages of 0.55 nm and 1.1 nm respectively. The RMSD in these complexes, AMP-12a and AMP-12b C-α atoms stabilize after around 60 ns, averaging around 0.55 nm. However, the RMSD in AMP-15 is the least among the AMPs, with an average of 0.2 nm, and AMP-6 has the highest, with an average of 0.8 nm. The results of RMSD in the C-α atom of squalene synthase bound to respective AMPs showed that the squalene synthase bound to AMP-3 has the lowest RMSD with an average of 0.6 nm and bound to AMP-6 has the highest RMSD with an average of 1.25 nm. The RMSD in the C-α atom of AMP-6 also has a high magnitude of deviations throughout the simulations. At the same time, the RMSD in the C-α atom of AMP-3 is reasonably stable, with an average of 0.5 nm.

**Root mean square fluctuation (RMSF) analysis.** The analysis of RMSF showed that the residues in the range 100–200 of histone demethylase bound to AMP-15 had the most

**Table 3. Effect of peptide rich extracts from *C. annum* and *S. tuberosum* on the Minimum Fungicidal Concentration (MFC) on *R. solani*.**

| Extracts | Concentrations mg/mL | | | | Positive Control |
|---|---|---|---|---|---|
| | 0 | 25 | 50 | 100 | |
| *C. annum* | ─ | ─ | ++ | + | + |
| *S. tuberosum* | ─ | ++ | + | + | + |

Note: (─) = Mycelial growth observed (MIC Negative); (+) = No mycelial growth observed (MIC Positive) and (++) = MFC Value.

**Table 4. Protein-peptide docking scores.**

| Peptide | Peptide Sequence | Docking Server | Histone demethylase | Sortin | Squalene |
|---|---|---|---|---|---|
| 1 | GFPFLLNGPDQDQGDFIMFG | HPEPDOCK | -224.484 | -211.439 | -211.576 |
| 2 | GFKGEQGVPQEMQNEQATIP | HPEPDOCK | -166.875 | -169.529 | -192.257 |
| 3 | RTCFCRRRLGRCDGGSF | HPEPDOCK | -217.423 | -221.035 | -225.028 |
| 4 | QICTNCCAGRKGCNYYSAD | HPEPDOCK | -207.735 | -206.433 | -218.308 |
| 5 | GICTNCCAGRKGCNYFSAD | HPEPDOCK | -183.883 | -196.685 | -200.798 |
| 6 | AGTNAVDLSVDQLCGVTSGRITTWNQLPATGR | HDOCK | -195.89 | -239.71 | -223.39 |
| 7 | ITYMSPDYAAPTLAGLDDATK | HPEPDOCK | -208.986 | -204.147 | -212.490 |
| 8 | RSASGTTELFTR | HPEPDOCK | -191.178 | -204.691 | -190.733 |
| 9 | MRFFATFFLLAMLVVATKMGPMRIAEARHCESLSHRFKGPCTRDSNCASVCETERFSGGNCHGFRRRCFCTKPC | HDOCK | -181.862 | -193.850 | -159.978 |
| 10 | MAKSLVSYTTHIALLLCFLLISSNEMQAAEGKLCRRKSKILGGSCLINRNCNKDCKEKEGAERGMCLKNDVFRHYCYCFHKCK | HDOCK | -258.35 | -254.91 | -279.57 |
| 11a | YASPSQGGQSQRSGGGGGGGGGGGGAGN | HDOCK | -188.03 | -174.34 | -190.83 |
| 11b | YGSPSQGGQSQRSGGGGGGGGGGGGGAGN | HDOCK | -199.1 | -201.68 | -210.99 |
| 12a | TAFYGPVGPPGRDSSGKG | HPEPDOCK | -208.866 | -225.324 | -194.177 |
| 12b | TAFYGPVGPRGRDSSGKG | HPEPDOCK | -218.532 | -225.324 | -218.721 |
| 13 | MKTIQGGQSATTALTMEVARVQA | HDOCK | -186.38 | -193.91 | -200.84 |
| 14 | LPSDATLVLDQTGKELDARL | HPEPDOCK | -174.706 | -161.697 | -163.215 |
| 15 | DICTCCAGTKGCNTTSANGAFICEGQSDPKKPKACPLNCDPHIAYA | HDOCK | -223.2 | -252.17 | -214.75 |

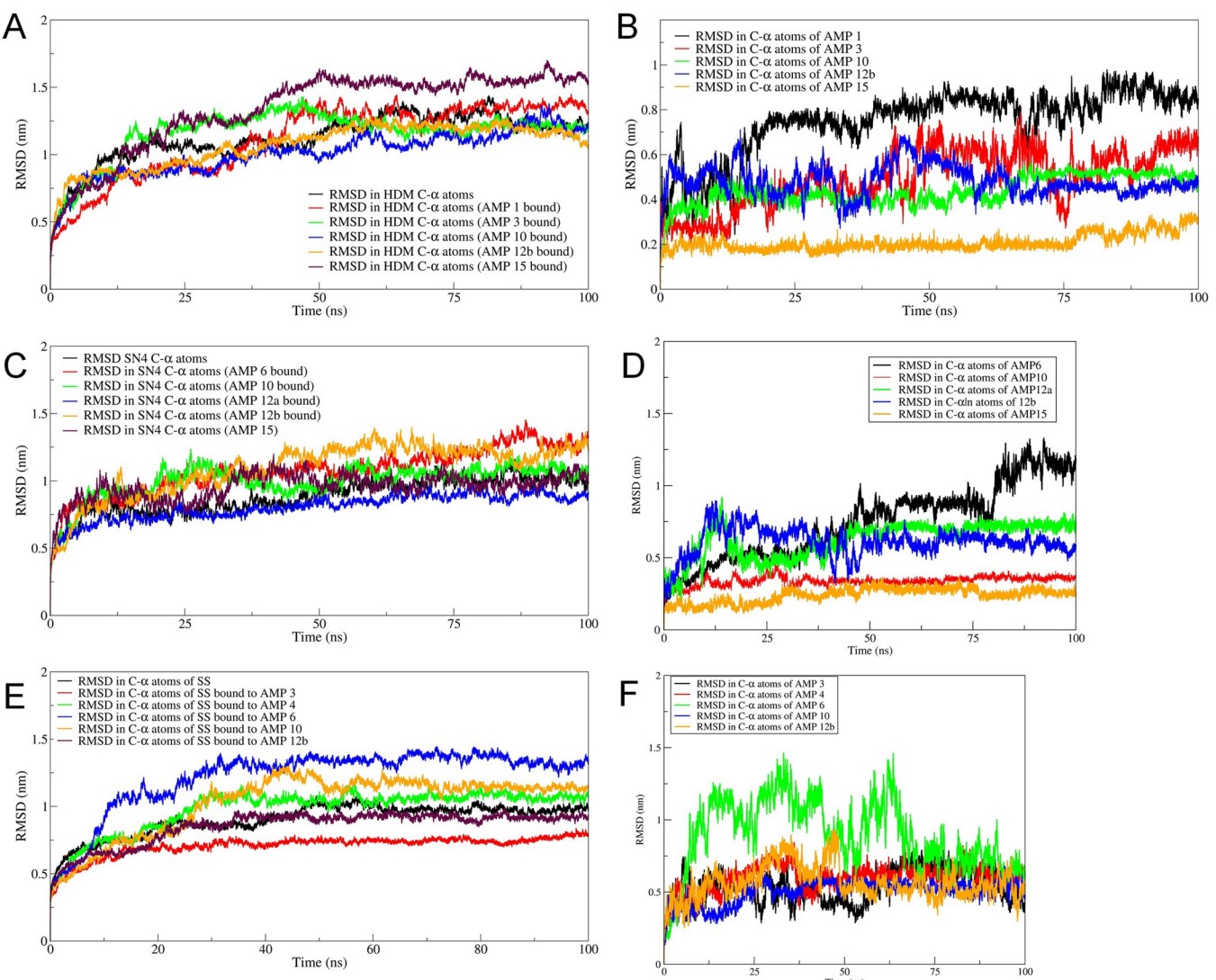

**Fig 2. Results of RMSD analysis.** RMSD in C-α atoms of A) histone demethylase, B) AMP bound to histone demethylase, C) Sortin, D) AMP bound to sortin, E) Squalene, and F) AMP bound to squalene.

significant fluctuations (**Fig 3**). Meanwhile, around 250–270 residues of histone demethylase bound to AMP-10 show slightly more significant fluctuations than those of other systems. The RMSF analysis of AMPs bound to histone demethylase showed the least fluctuations in AMP-15 residues, while AMP-10 showed the highest fluctuations in magnitude. The RMSF analysis of sortin showed that almost all the residues of sortin bound to AMP-12b had a higher magnitude of fluctuations compared to other systems. While the RMSF in bare sortin and sortin bound to other AMPs showed almost analogues RMSF. The RMSF in residues of AMP bound to sortin showed the least fluctuations in AMP-15 and the highest fluctuations in AMP-6. The least fluctuations were seen in the residues of squalene bound to AMP-3, while slightly higher fluctuations were seen in the squalene residues bound to AMP-4 and AMP-6. The RMSF in AMP residues in AMPs bound to squalene systems showed the highest fluctuations in AMP-6 and the lowest fluctuations in AMP-10.

**Radius of gyration (Rg) analysis.** The Rg analysis of individual chains of each protein and each bound AMPs is shown in **Fig 4**. The Rg in histone demethylase bound to AMP-15

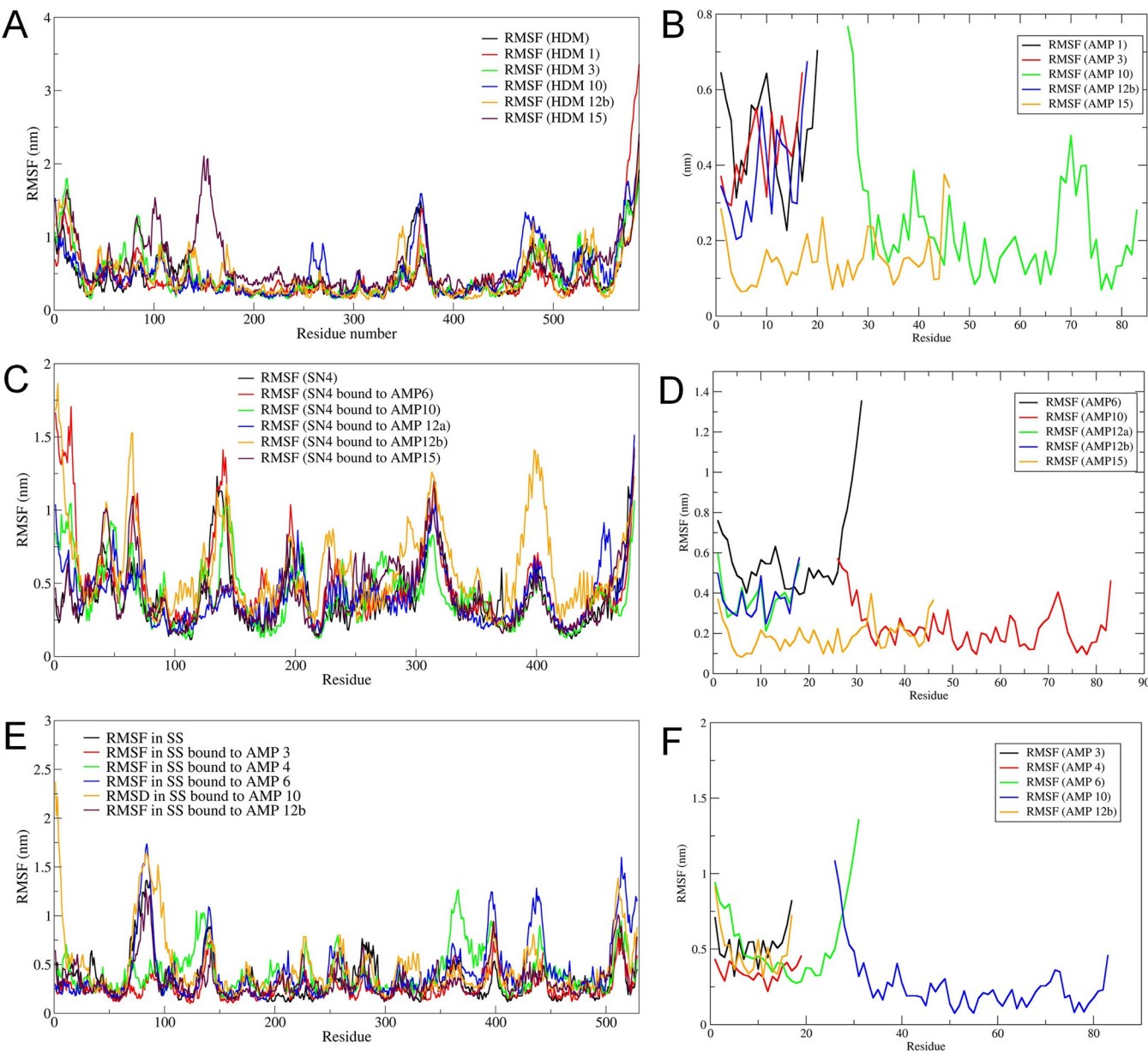

**Fig 3. Results of RMSF analysis.** RMSF in residues of A) histone demethylase, B) AMP bound to histone demethylase, C) Sortin, D) AMP bound to sortin, E) Squalene, and F) AMP bound to squalene.

showed a slightly larger deviation (**Fig 4A**), while the AMP-15 chain bound to histone demethylase showed the least deviations (**Fig 4B**). The Rg in histone demethylase bound to AMP-12b is lowest compared to other systems, while Rg in AMP-12b is seen stabilized after around 60 ns after initial deviations. Similarly, the sortin bound to AMP-10 showed the largest deviations, though the Rg in individual chains of AMP-10 is slightly higher but stable throughout the simulation period. The least deviations in Rg are seen in sortin bound to AMP-15 and the least in the individual chain of AMP-15. The Rg in squalene bound to AMP-3 is lowest while that bound to AMP-4 is highest. However, the Rg in individual chains of AMPs in systems bound to squalene showed the lowest Rg in AMP-12b and the highest in AMP-4.

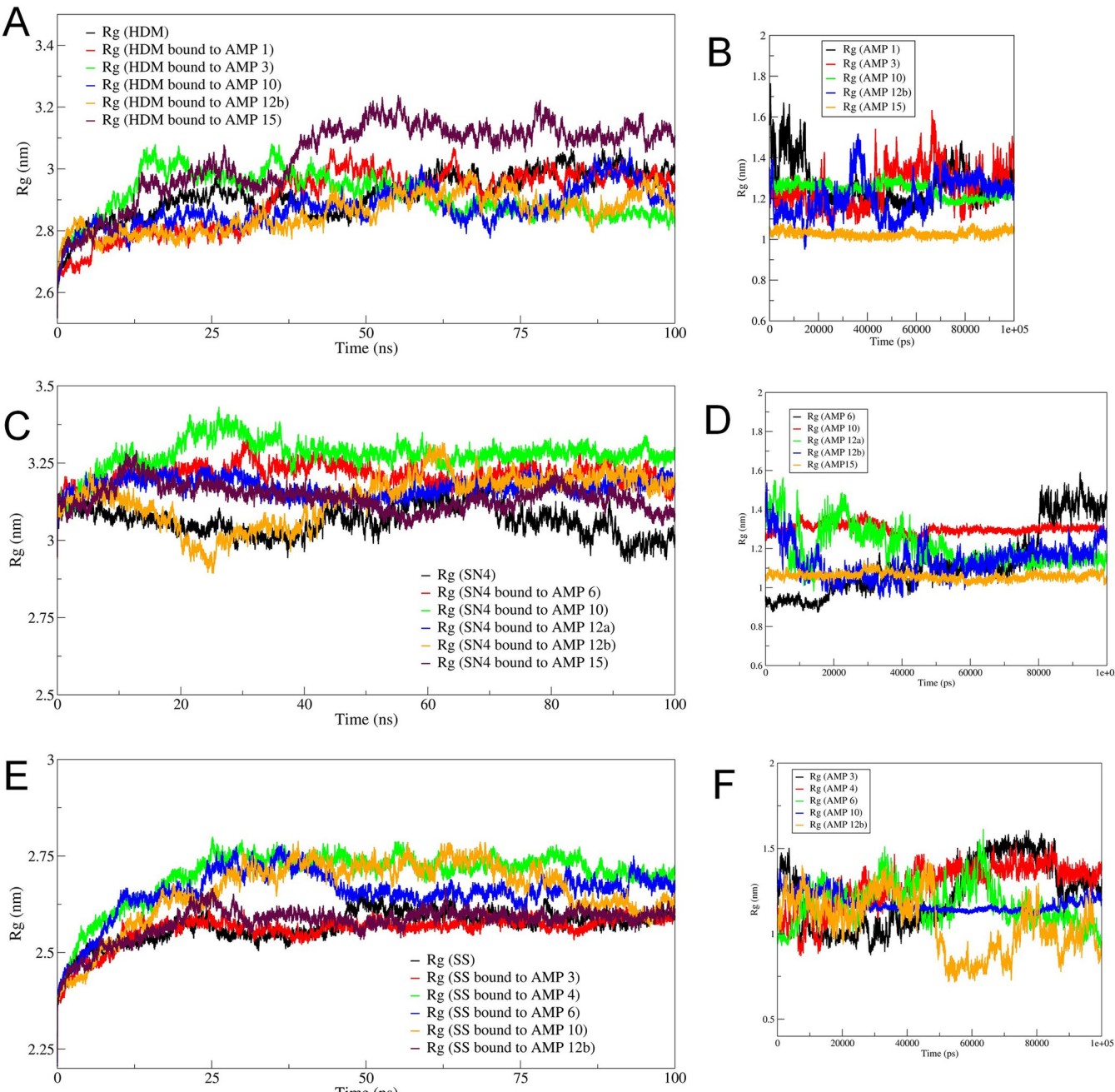

**Fig 4. Results of radius of gyration.** Rg in A) histone demethylase, B) AMP bound to histone demethylase, C) Sortin, D) AMP bound to sortin, E) Squalene, and F) AMP bound to squalene.

**Hydrogen bond analysis.** The results of hydrogen bond analysis between histone demethylase and AMPs showed that the maximum hydrogen bonds are formed between the histone demethylase and AMP-3 throughout the simulation period, reaching a maximum of around 19 hydrogen bonds (**Fig 5A**). A number of hydrogen bonds are formed with AMP-15. AMP-1 forms an average of 7 hydrogen bonds after around 20 ns, while AMP-12b forms 10 hydrogen bonds after around 80 ns simulation period. AMP-10 forms an average of 3

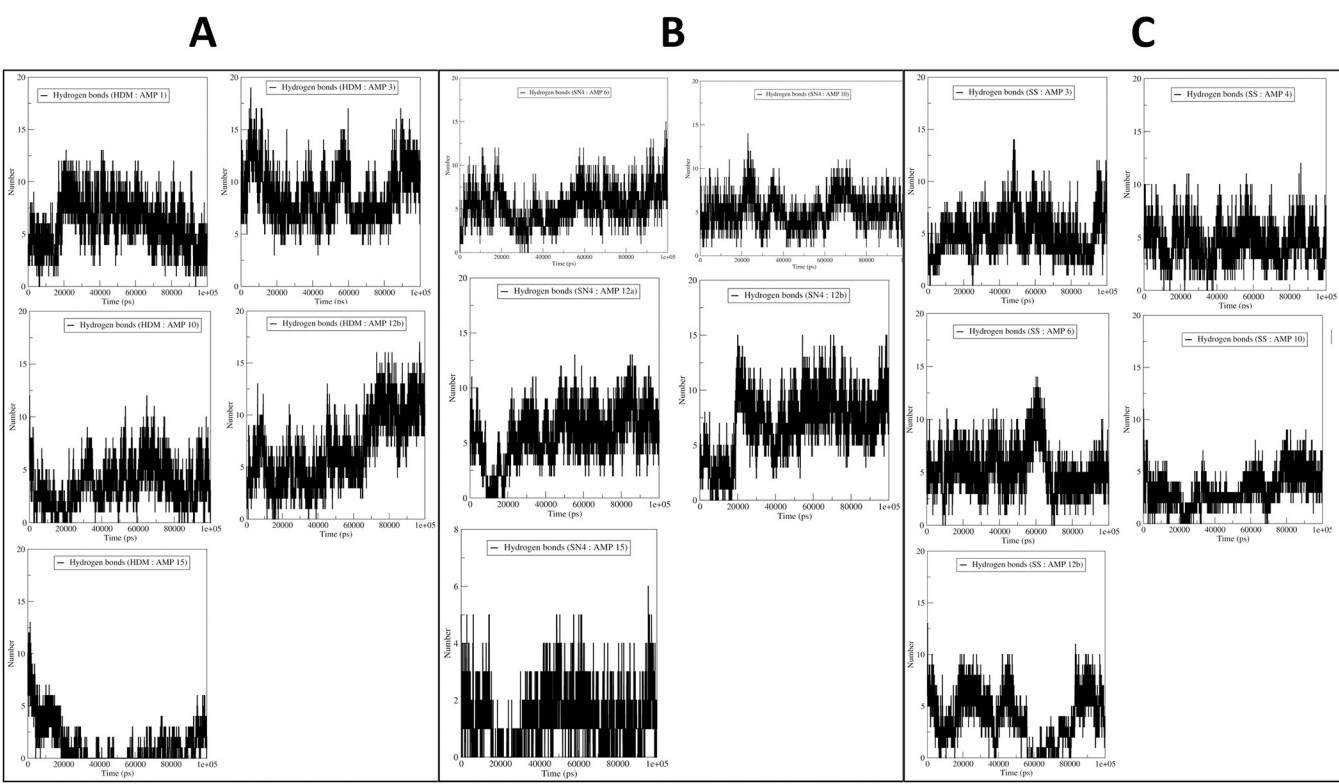

**Fig 5.** Hydrogen bond analysis for (A) histone demethylase and AMPs, (B) sortin and AMPs and (C) squalene and AMPs.

hydrogen bonds throughout the simulation period. In the case of sortin complexes, the AMP-12b forms the maximum hydrogen bonds amongst all the systems, with an average of 10 hydrogen bonds after a simulation period of around 20 ns (**Fig 5B**).

Further, AMP-6, AMP-10, and AMP-12a form an average of around 5 hydrogen bonds. AMP-15 forms the least hydrogen bonds, with an average of around 2. In the case of squalene complexes, the AMPs, namely AMP-3, AMP-4, and AMP-6, formed an average of around 5 hydrogen bonds throughout the simulation (**Fig 5C**). The hydrogen bonds were seen less in number with AMP-10 and AMP-12b.

**PCA and Gibb's free energy analysis.** Principal component analysis (PCA) was performed to understand each complex's motion path, and the corresponding system's stability was studied through Gibb's free energy landscape analysis. The bare Histone demethylase showed unique conformations, some occupying the lower energy basin in the range -20 to -10 kJ mol$^{-1}$ on PC1 and 0 to -10 kJ mol$^{-1}$ on PC2 (**Fig 6**). The majority of the conformations of histone demethylase bound to AMP1 occupy the energy basin with slightly higher energy in the range of 10 to 20 kJ mol$^{-1}$ on PC1 and 5 to -5 on PC2. The histone demethylase bound to AMP3 also showed the majority of conformations with slightly higher energy. The histone demethylase bound to AMP10 showed unique conformations with the lowest energy in the energy basin 0 to -25 kJ mol$^{-1}$ on PC1 and 0 to -25 kJ mol$^{-1}$ on PC2. The AMP12b bound system showed many conformations occupying a lower energy basin similar to bare histone demethylase. The histone demethylase bound to AMP15 has high energy conformations from 0 to 25 kJ mol$^{-1}$ on PC1 and -20 to -40 on PC2. The bare sortin showed many conformations occupying the lowest energy basin with energy in the range -10 to -15 kJ mol$^{-1}$ on PC1 and -5 to -10 kJ mol$^{-1}$ on PC2 (**Fig 7**). The complexes with AMP6 and AMP12a showed many

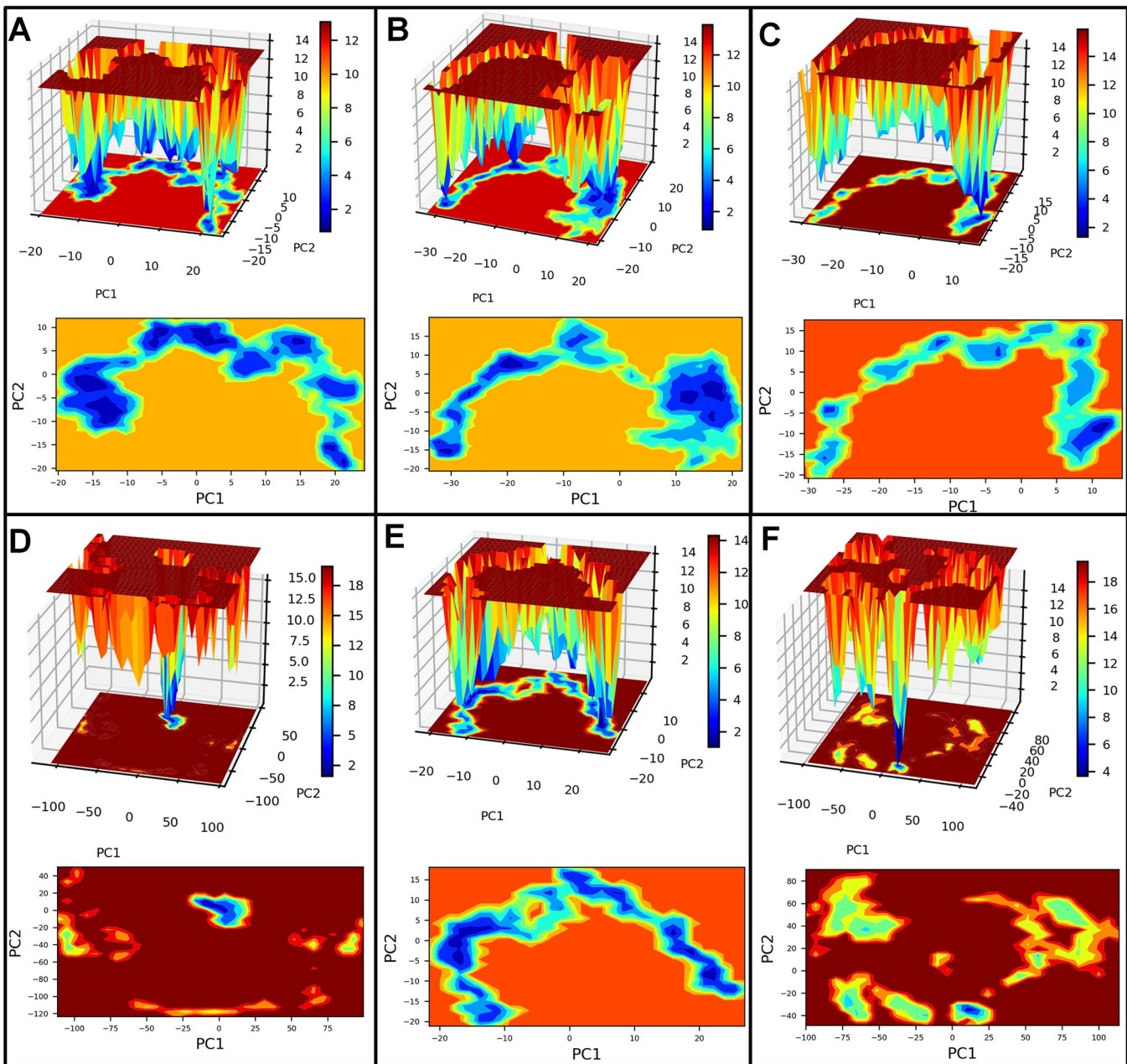

**Fig 6. Gibb's free energy analysis for histone demethylase complexes.** A) Histone demethylase, B) Histone demethylase-AMP1 complex, C) Histone demethylase-AMP3 complex, D) Histone demethylase-AMP10 complex, E) Histone demethylase-AMP12b complex, and F) Histone demethylase-AMP15 complex.

conformations occupy the lowest energy basin in the range -10 to -20 kJ mol$^{-1}$ on PC1 and 0 to -10 kJ mol$^{-1}$ on PC2. The complex with AMP10 occupied a unique low-energy basin on PC2 in the range of -5 to -10 kJ mol$^{-1}$ on PC2. For complex with AMP12b, few conformations were seen in the low energy basin, while in complex with AMP15, many conformations were seen occupying the lowest energy basin with energy -10 to -30 kJ mol$^{-1}$ on PC1 and 0 to -15 kJ mol$^{-1}$ on PC2. The bare squalene synthase showed the largest energy basin with slightly high energy in the range of 5 to 10 kJ mol$^{-1}$ on PC1 and 0 to 5 kJ mol$^{-1}$ on PC2 (**Fig 8**). The complexes with

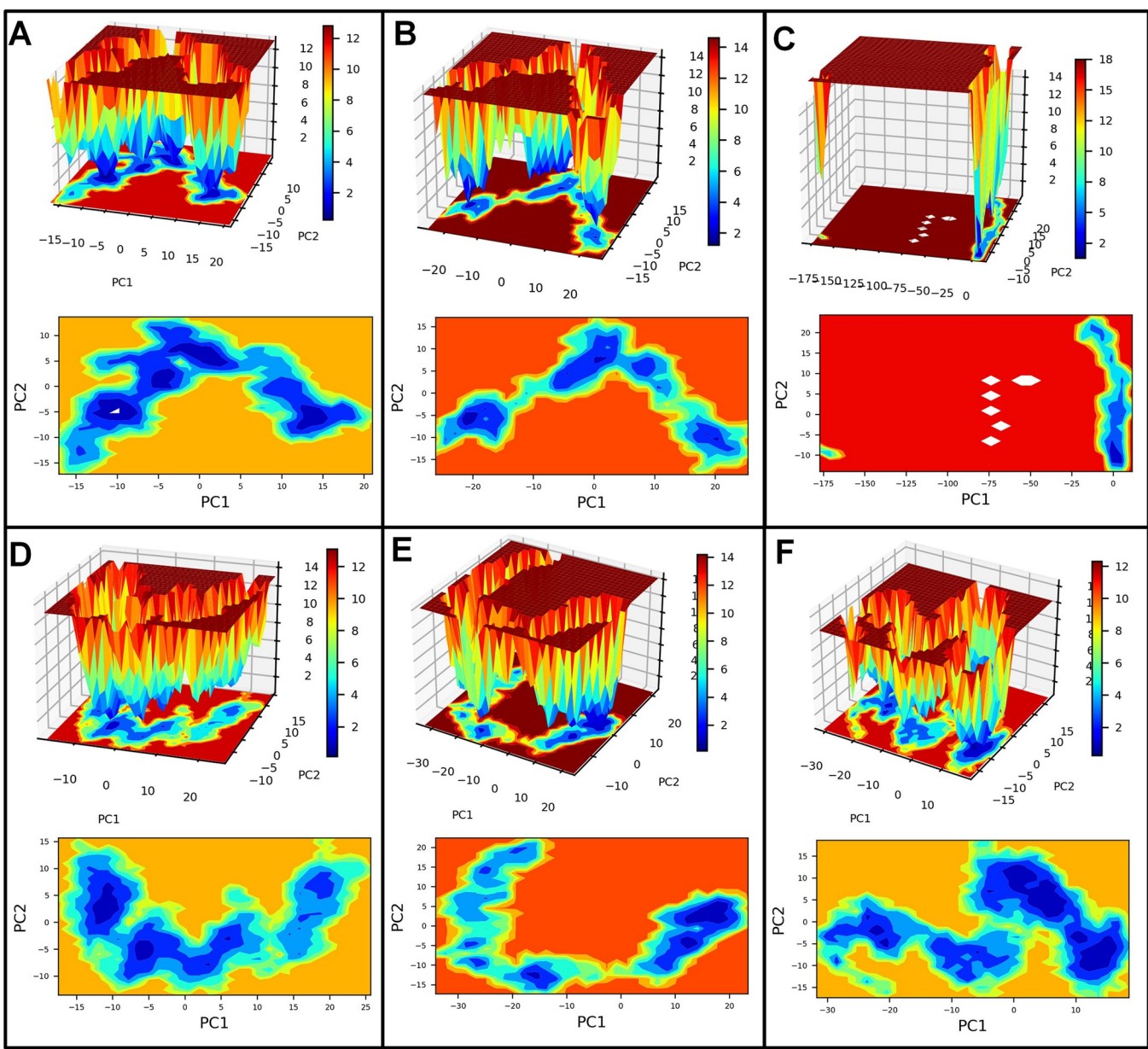

**Fig 7. Gibb's free energy analysis for sortin complexes.** A) Sortin, B) Sortin AMP6 complex, C) Sortin AMP10 complex, D) Sortin AMP12a complex, E) Sortin AMP12b complex, and F) Sortin AMP15 complex.

AMP3 and AMP10 similarly have slightly higher energy basins where most conformations were occupied. The complex with AMP4 showed conformations occupying the lower energy basin in the range -5 to -15 kJ mol⁻¹ on PC1 and 0 tp -10 kJ mol⁻¹ on PC2. The complex with AMP6 and 12b showed few conformations occupying lower energy basins.

**Definition of secondary structure of proteins (DSSP) analysis.** The secondary structural changes during MD simulation were evaluated by defining the secondary structure of proteins (DSSP) analysis. The results showed that the histone demethylase bound to AMP1 has a secondary structure change in 425 onward residues 425 and residues in the range 12–16 in AMP1 (**Fig 9**). Similar structural changes in histone demethylase were observed in histone

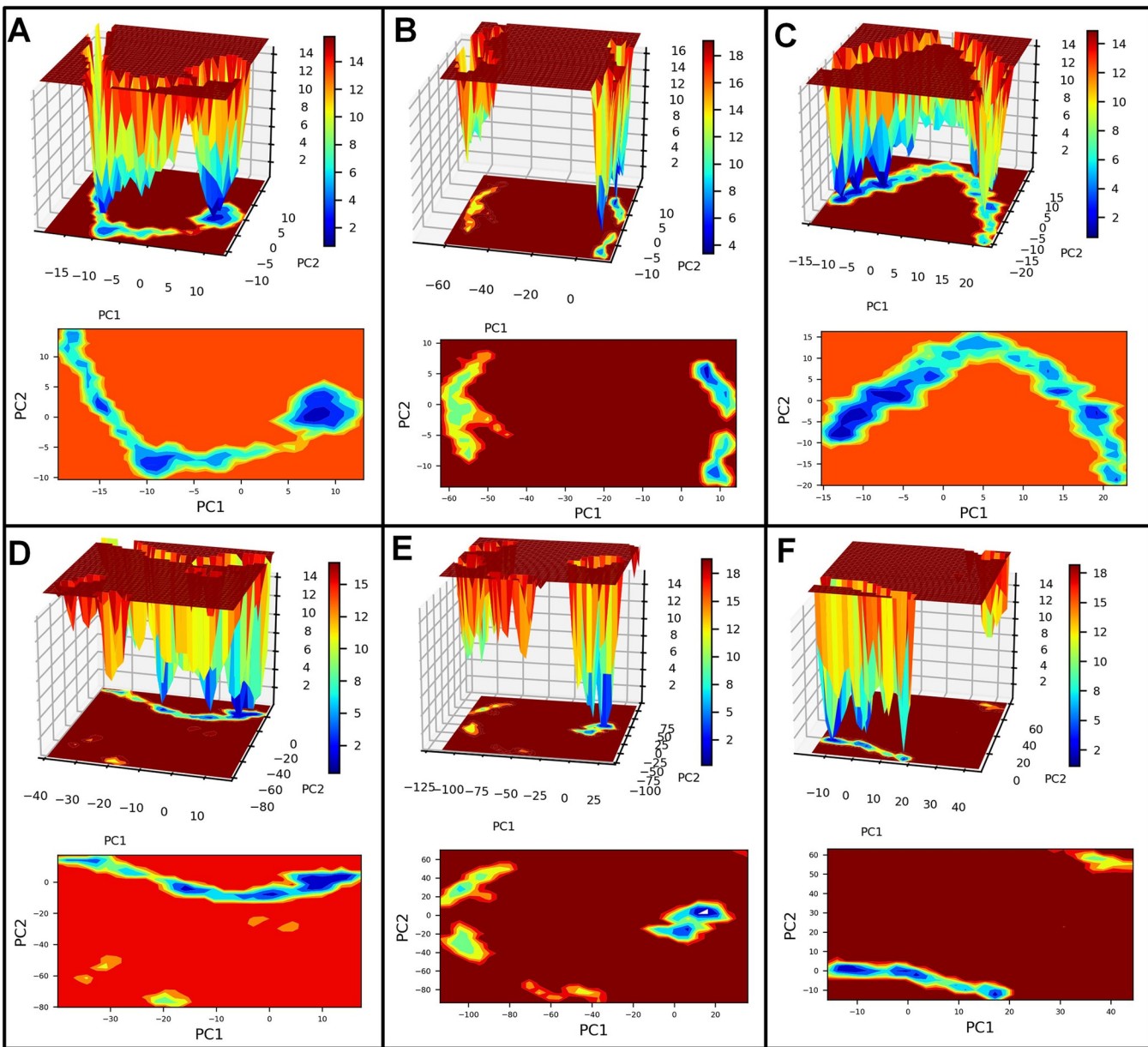

**Fig 8. Gibb's free energy analysis for squalene complexes.** A) Squalene, B) Squalene AMP3 complex, C) Squalene AMP4 complex, D) Squalene AMP6 complex, E) Squalene AMP10 complex, F) Squalene AMP12b complex.

demethylase bound to AMP3, while the residues 8–14 in AMP3 had secondary structural changes. The histone demethylase in the remaining complexes also showed similar secondary structural changes in the 425 onward residues. The bound AMP10 showed secondary structural changes in the residues 10–22, 32–37, and 41–51. The residues 12–15 bound AMP12a found the major secondary structural change. Many residues in AMP15 were seen as having secondary structural changes. In the case of sortin, in all the complexes, the secondary structural changes were seen in the residues ranging from 250–300 and 350–400 (**Fig 10**). AMP6 bound to sortin showed major secondary structural changes in the residues 11–20. AMP10 bound to sortin showed major secondary structural changes in the residues ranging from 2–5

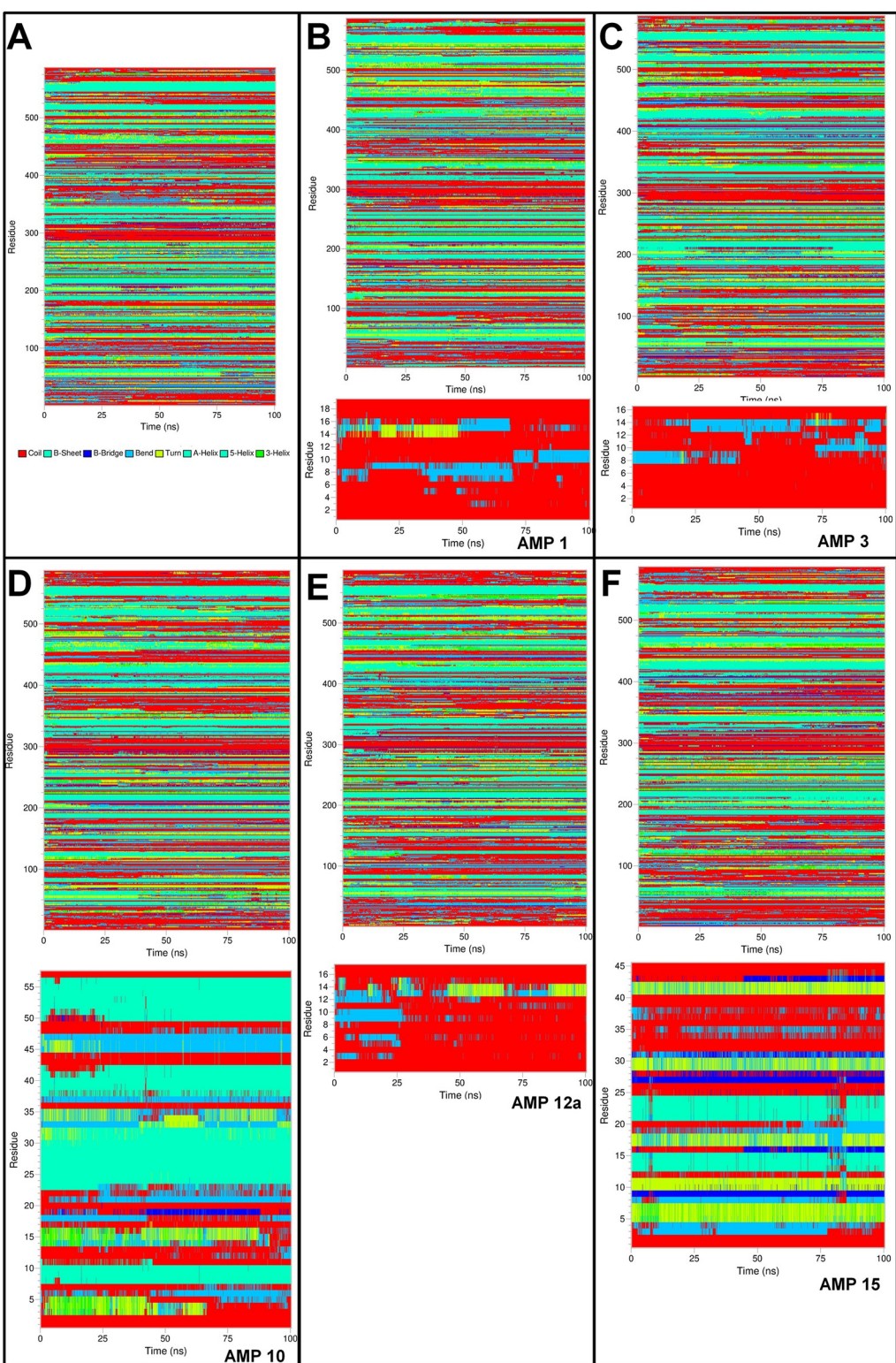

**Fig 9. DSSP analysis of histone demethylase and bound AMPs.** A) Histone demethylase, B) Histone demethylase-AMP1 complex, C) Histone demethylase-AMP3 complex, D) Histone demethylase-AMP10 complex, E) Histone demethylase-AMP12b complex, and F) Histone demethylase-AMP15 complex.

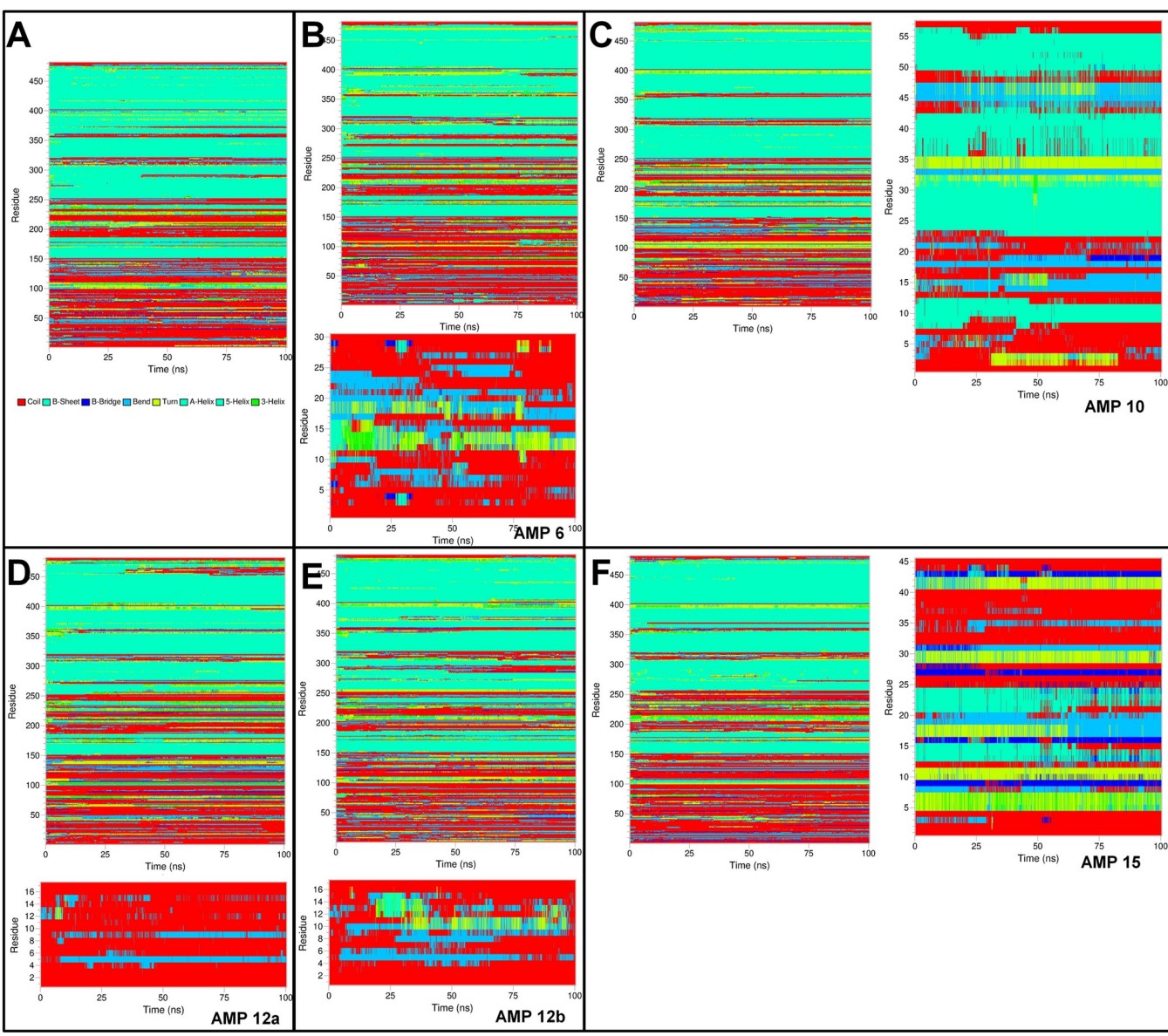

**Fig 10. DSSP analysis of Sortin and bound AMPs.** A) Sortin, B) Sortin AMP6 complex, C) Sortin AMP10 complex, D) Sortin AMP12a complex, E) Sortin AMP12b complex, and F) Sortin AMP15 complex.

and 45–50. The AMP12a showed very few secondary structural changes, while AMP12b showed major changes in the residues ranging from 8–12. In AMP15, secondary structural changes were seen in the residues ranging from 5 to 25. In the case of squalene synthase, the secondary structural changes were seen in the residues 200–350 in all the complexes (**Fig 11**). AMP3 bound to squalene synthase showed secondary structural changes in the residues in the range 8–12. Major structural changes in AMP4 were seen in the residues ranging from 14–16. AMP6 bound to squalene synthase showed secondary structural changes in the residues ranging from 2–7 and 10–15. The major structural changes in AMP10 were seen in the residues ranging from 5–23. In AMP12b, structural changes were seen in the residues ranging from 4 to 12.

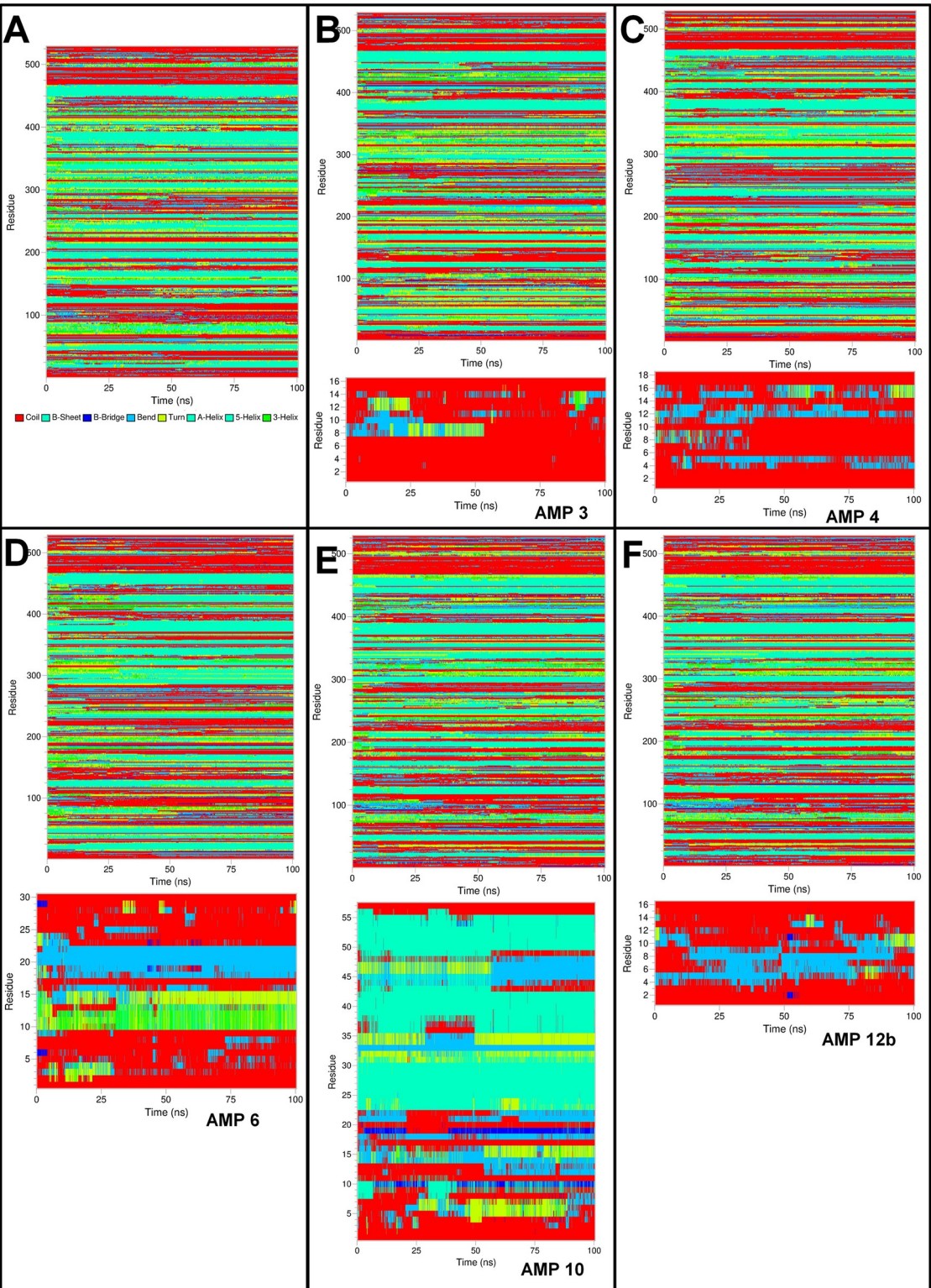

**Fig 11. DSSP analysis of Squalene and bound AMPs.** A) Squalene, B) Squalene AMP3 complex, C) Squalene AMP4 complex, D) Squalene AMP6 complex, E) Squalene AMP10 complex, F) Squalene AMP12b complex.

**Dynamic cross-correlation matrix (DCCM) analysis.** The time-correlated information of inter-chain and intra-chain residue to residue contacts between the proteins and AMP chains was studied through DCC analysis. The comprehensive residue-residue crosswalk and the dynamic cross-correlation matrix (DCCM) were constructed for each complex. The independent motions and residue-residue crosswalk were seen between the histone demethylase and AMP chains (**Fig 12**). The color gradient range is from blue (negative correlation, less likely) to red (positive correlation, more evident), corresponding to the correlation coefficients -1 and +1, respectively, and lighter shades indicate weaker correlations. The complexes of histone demethylase with AMP1 and AMP3 showed moderately positively correlated residue-residue crosswalks. The complexes with AMP10 and AMP15 showed strong negatively correlated residue-residue crosswalks. The complex with AMP12b showed slightly better positively correlated residue-residue crosswalks. Sortin with 481 residues showed no correlation with AMP6 (**Fig 13**). However, the residues up to 250 from the sortin chain showed stronger positively correlated residue-residue crosswalks with AMP10 and AMP12a. Few positively correlated residue-residue crosswalks were seen for other AMPs, namely AMP12b and AMP15. Squalene synthase complexes AMP6 and AMP10 showed strong negatively correlated residue-residue crosswalks (**Fig 14**). AMP3 and AMP4 showed few moderate positively correlated residue-residue crosswalks, but the unique residue-residue crosswalks in the squalene synthase chain were seen lost in these complexes. AMP12b showed a few moderate positively correlated residue-residue crosswalks with intact residue-residue crosswalks in the squalene synthase chain.

**MM-PBSA calculations.** The trajectories extracted at each 500 ps between the simulation periods of 75 ns to 100 ns were subjected to MM-PBSA calculation. The interaction energies such as van der Waal, electrostatic, polar solvation, SASA, and binding energy ($\Delta G_{binding}$) between individual proteins and AMP were estimated. The results of MM-PBSA calculations are given in **Table 5**. The results show that the AMP10 has the most favourable binding energy of -492.099 kJ mol$^{-1}$. AMP3 has favourable binding energy but far more (-58.379 kJ mol$^{-1}$) than AMP10. Other AMPs bound to histone demethylase show unfavourable binding energy. In the case of sortin complexes, AMP10 showed the most favourable binding energy, five times less than AMP15. Other AMPs bound to sortin showed slightly unfavourable binding energies. In the case of squalene synthase, AMP6 is the only AMP which showed the most favourable binding energy, while other AMPs showed unfavourable binding energy.

## Discussion

Multiple antimicrobial peptides from various species of the family of Solanaceae have been reported including *C. annum* and *S. tuberosum* (Afroz *et al.*, 2020 [8]). In this study, we provide the initial evidence that extracts rich in peptides from *C. annum* and *S. tuberosum* effectively inhibit the rice sheath blight pathogen, *R. solani*, in laboratory settings. Our results demonstrated that the inhibition zone was the highest at 100 mg/ml doses for both *S. tuberosum* and *C. annum* peptide extract. Specifically, *S. tuberosum* inhibited fungal growth by 46.0 mm, while *C. annum* peptide extract inhibited fungal growth by 40.67 mm (**Table 1**). On the other hand, the lowest inhibition zone was observed at 25 mg/ml, with *S. tuberosum* inhibiting *R. solani* growth by 36.33 mm and *C. annum* growth by 32.67 mm. The negative control (aquous DMSO) did not exhibit any impact on the inhibition zone, whereas the positive control (Azostrobin) inhibited growth by 38.67 mm. Statistical analysis revealed that the mean zone of inhibition varied significantly with different doses in both extracts (P < 0.05). However, in the case of *C. annum* extract, there were no statistically significant differences in the mean zone of inhibition at 50 mg/ml, 100 mg/ml, and the positive control. Similarly, the mean difference in *S. tuberosum* extract at 25 mg/ml, 50 mg/ml, and the positive control was not considered

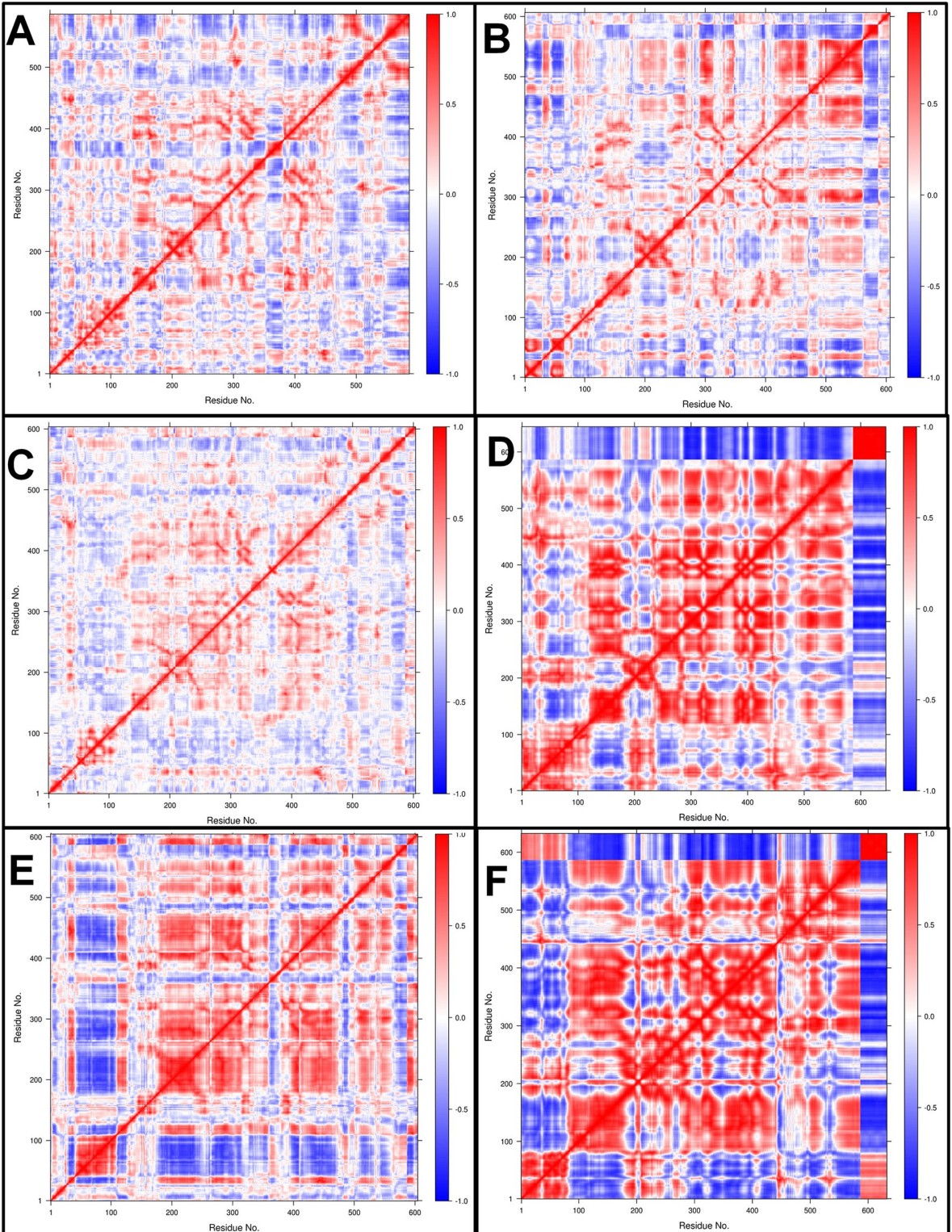

**Fig 12. DSSP analysis of Histone demethylase AMP complexes.** A) Histone demethylase, B) Histone demethylase-AMP1 complex, C) Histone demethylase-AMP3 complex, D) Histone demethylase-AMP10 complex, E) Histone demethylase-AMP12b complex, and F) Histone demethylase-AMP15 complex.

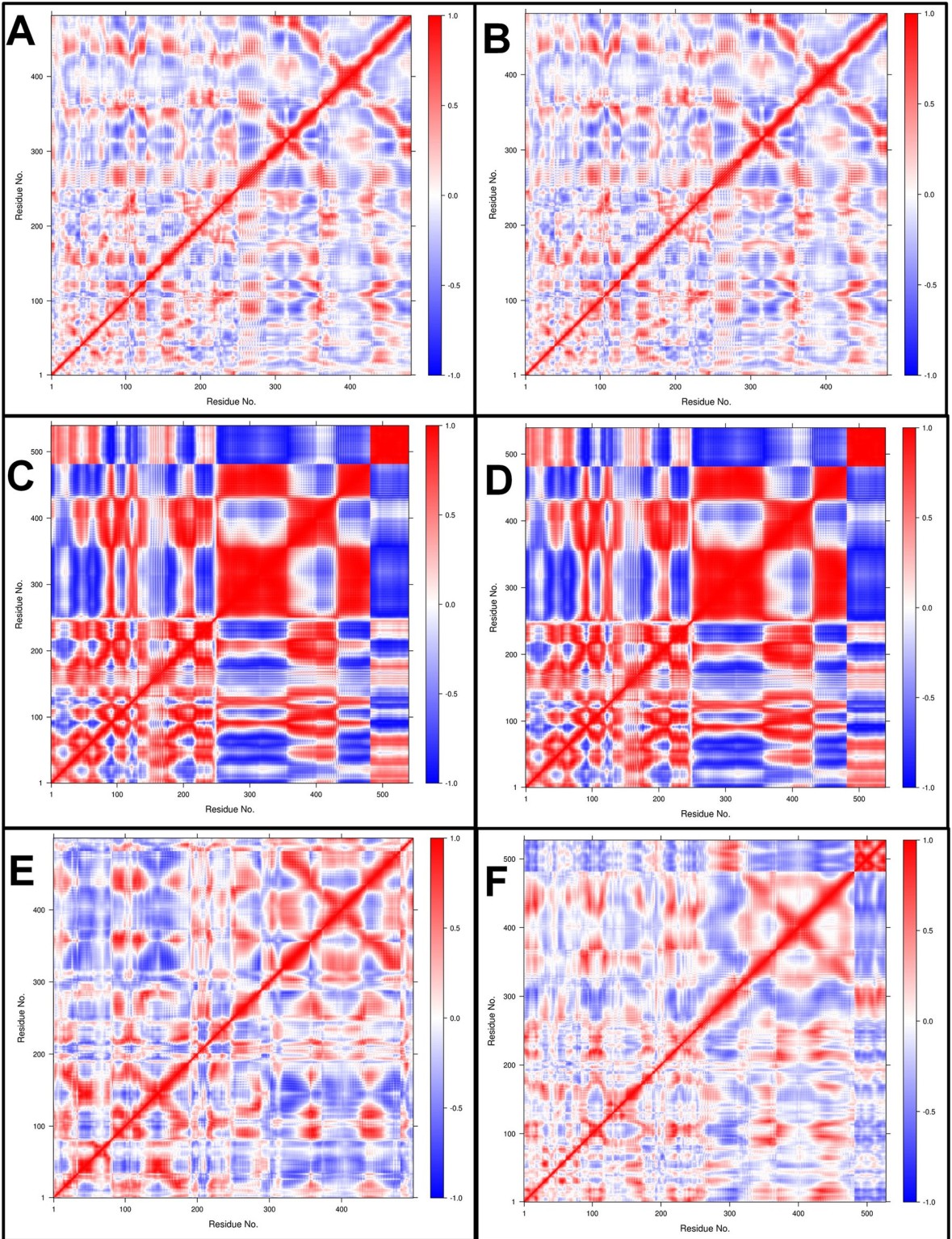

**Fig 13. DSSP analysis of Sortin AMP complexes.** A) Sortin, B) Sortin AMP6 complex, C) Sortin AMP10 complex, D) Sortin AMP12a complex, E) Sortin AMP12b complex, and F) Sortin AMP15 complex.

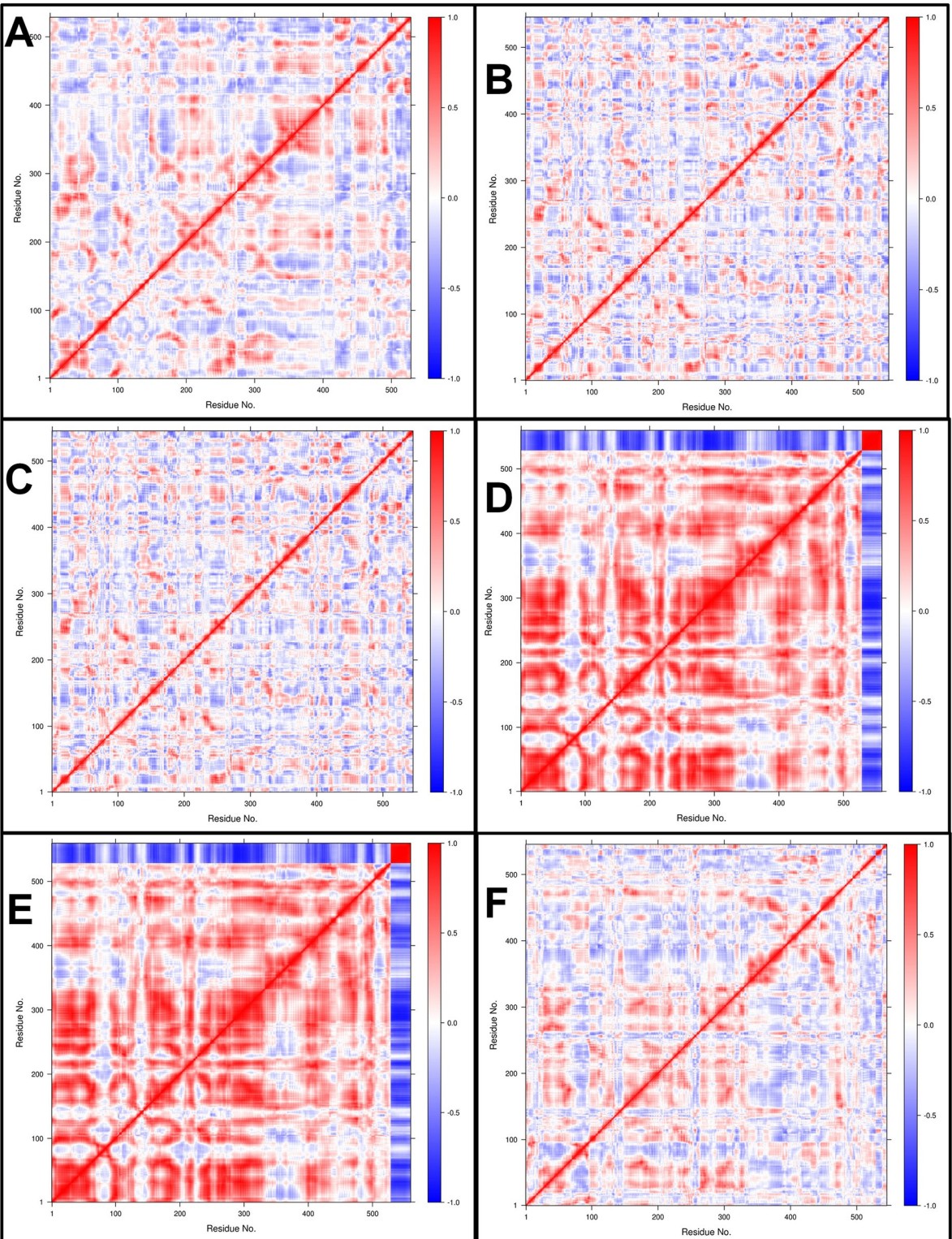

**Fig 14. DSSP analysis of Squalene AMP complexes.** A) Squalene, B) Squalene AMP3 complex, C) Squalene AMP4 complex, D) Squalene AMP6 complex, E) Squalene AMP10 complex, F) Squalene AMP12b complex.

**Table 5. Results of MM-PBSA calculations.**

| | van der Waal energy (kJ mol$^{-1}$) | Electrostatic energy (kJ mol$^{-1}$) | Polar solvation energy (kJ mol$^{-1}$) | Solvent accessible surface area energy (kJ mol$^{-1}$) | Binding energy ($\Delta G_{binding}$) (kJ mol$^{-1}$) |
|---|---|---|---|---|---|
| **Histone demethylase** | | | | | |
| AMP1 | -295.157 (1.647) | -162.168 (2.620) | 539.399 (7.520) | -40.584 (0.188) | 41.412 (5.797) |
| AMP3 | -264.162 (1.834) | -551.971 (2.084) | 798.013 (6.520) | -40.072 (0.299) | -58.379 (4.205) |
| AMP10 | -165.517 (4.030) | -934.674 (16.429) | 633.594 (16.894) | -25.887 (0.588) | **-492.099 (8.583)** |
| AMP12b | -411.882 (1.279) | -322.217 (1.857) | 828.610 (5.137) | -57.340 (0.122) | 36.948 (3.736) |
| AMP15 | -24.107 (1.894) | -6.492 (1.655) | 69.297 (5.708) | -3.324 (0.281) | 35.730 (5.107) |
| **Sortin** | | | | | |
| AMP6 | -359.262 (1.703) | -54.589 (1.710) | 479.249 (3.605) | -50.976 (0.201) | 14.286 (3.181) |
| AMP10 | -294.172 (4.116) | -980.412 (8.666) | 741.416 (9.982) | -41.159 (0.514) | **-573.159 (6.499)** |
| AMP12a | -298.651 (1.226) | -441.355 (1.393) | 822.073 (5.004) | -44.233 (0.131) | 37.582 (3.374) |
| AMP12b | -302.775 (2.025) | -580.473 (1.554) | 1087.148 (4.199) | -50.009 (0.220) | 154.215 (3.850) |
| AMP15 | -171.112 (1.420) | -147.763 (1.546) | 231.204 (4.997) | -23.094 (0.198) | -110.914 (4.205) |
| **Squalene synthase** | | | | | |
| AMP3 | -251.265 (1.835) | 543.448 (2.367) | 538.975 (7.374) | -36.364 (0.248) | 795.692 (5.701) |
| AMP4 | -296.434 (1.566) | 58.454 (2.741) | 565.741 (7.833) | -42.842 (0.173) | 284.889 (6.324) |
| AMP6 | -193.454 (2.254) | -240.635 (2.877) | 265.870 (5.873) | -25.370 (0.298) | **-193.766 (3.188)** |
| AMP10 | -120.579 (1.632) | 797.951 (2.200) | 557.979 (4.165) | -22.114 (0.207) | 1213.016 (3.853) |
| AMP12b | -143.990 (1.265) | 157.671 (1.668) | 405.012 (5.616) | -21.886 (0.179) | 396.944 (5.125) |

significant. These findings may be attributed to factors such as the quantity of substance applied in the agar plate's hole, the thickness of the extract, and its ability to penetrate the pore spaces of the agar media (Horváth *et al.*, 2016 [54]).

Our results (Table 1) demonstrated that mycelial growth was prevented at a minimum concentration of 50 mg/ml of *C. annum* extract and 25 mg/ml of *S. tuberosum* extract. The inhibitory effect of *C. annum* extract could be attributed to the presence of antimicrobial peptides in the leaf, and its significance in this study is consistent with Dev and Venu (2016) [10] and Maracahipes *et al.* (2019) [11]. In the report of both groups, a significant inhibitory effects of the peptide leaf extracts on fungal growth and hyphae proliferation were demonstrated. The inhibitory effect of *S. tuberosum* could also be attributed to the presence of low molecular weight potato peptides such as defensins, thionin, and snakins, which have been shown in several studies to have antifungal effects on several plant pathogenic fungi [49–51].

In our present study, the Minimum Inhibitory Concentration (MIC) and Minimum Bactericidal Concentration (MBC) values (Tables 2 and 3) were determined to be 50 mg/ml for the *C. annum* extract and 25 mg/ml for the *S. tuberosum* extract. This elucidates the fungistatic

and fungicidal concentrations against *R. solani*, which were found to be 50 mg/ml in the peptide-rich leaf extract of *C. annum* and 25 mg/ml in the peptide-rich tuber extract of *S. tuberosum*. Our findings are in agreement with the MBC of 64 g/ml reported by [52] in the cases of *Fusarium solani* and *F. oxysporum*. Lee et al [53] reported a MIC/MBC concentration of *C. annum* extract greater than 100 g/ml on *R. solani*, whereas [50] reported 100 M as MIC/MBC on *R. solani*. These values differ from those reported in this study and may be due to genetic variability (strain) of the pathogen, pH of the media, and incubation temperature [54]. The antimicrobial CaAMP1 protein in the leaf extract may be responsible for the inhibitory effect of *C. annum* extract on *R. solani* by inhibiting fungal spore germination and hyphal growth, while that of *S. tuberosum* may be due to a trypsin inhibitor. In silico screening has successfully been utilized for identification of potential fungicide against destructive rice blast fungus *Magnaporthe oryzae* [55,56].

Our protein-peptide docking studies revealed that AMP1, AMP3, AMP10, AMP12b, and AMP15 had the most favourable binding free energy against histone demethylase. Similarly, AMP6, AMP10, AMP12a, AMP12b, and AMP15 gave the most favourable binding energy against sortin. Against squalene synthase, the AMP3, AMP4, AMP6, AMP10, and AMP12a resulted in the most favourable binding energy. Most of these AMPs had favourable binding energy against a couple of target proteins except AMP1 and AMP4, which have favourable binding energy against histone demethylase and squalene synthase, respectively. All these complexes, along with bare proteins, were subjected to MDS studies to gain deeper insights into the stability of the respective complexes compared to the bare proteins.

The RMSD in the C-α atom of histone demethylase revealed that AMP10 stabilized the system more prominently compared to other AMPs. Mainly, AMP15 induced a lot of deviations in histone demethylase compared to bare protein. The RMSD in AMP10 atoms also suggest convergence after around 60 ns to a stable RMSD. AMP10 could be the most favourable AMP with the most favourable interactions with histone demethylase. In the case of sortin, AMP12a had the least RMSD in C-α atoms of sortin. However, the RMSD in AMP12a atoms was slightly higher than AMP12b, AMP10, and AMP15. Sortin complexes with AMP10 and AMP15 showed reasonable stabilization in terms of RMSD in C-α atoms of sortin and peptide atoms. Squalene synthase AMP3 showed the lowest RMSD in protein C-α and peptide atoms. The RMSF analysis in histone demethylase complexes suggested that AMP1 and AMP3 had the least fluctuations while AMP15-bound histone demethylase undergoes the most fluctuations. Possibly, AMP3 was most favourably bound and interacted with histone demethylase. In the case of sortin, AMP10 showed uniform and fewer fluctuations than other AMP-bound complexes. AMP3-bound squalene synthase has the least fluctuations and may be the most favourable against squalene synthase.

The radius of gyration (Rg) analysis revealed distinct characteristics among the complexes studied. Notably, the Rg for the histone demethylase chain bound to AMP3 exhibited the lowest value compared to all other histone demethylase complexes, while the Rg for the AMP15 chain was also notably low. Moreover, the Rg of the AMP10 chain stabilized around 60 ns, indicating its reasonable stability and compactness. Similarly, for sortin, both the AMP15 bound sortin chain and the individual AMP15 chain demonstrated the lowest Rg values, suggesting the stability and compactness of this complex. Regarding squalene synthase, the Rg for the squalene synthase chain bound to AMP3 was the lowest observed, while the individual AMP10 chain, when bound to squalene synthase, also exhibited the lowest Rg. These observations imply that these complexes are likely the most stable and compact among the studied configurations.

Our hydrogen bond analysis revealed notable differences among the complexes studied. Specifically, AMP3 exhibited the highest number of inter-chain hydrogen bonds with histone

demethylase, while AMP15 formed the fewest. Given the significance of hydrogen bond formation as a non-bonded interaction, the histone demethylase-AMP3 complex appears particularly favorable. Similarly, in the case of sortin, a considerable number of inter-chain hydrogen bonds were observed between sortin and AMP12b, indicating stability and favorable binding affinity. Conversely, AMP15 formed the least hydrogen bonds with sortin, potentially suggesting a somewhat lower binding affinity. Moreover, the AMP6-bound squalene synthase complex displayed a consistent formation of hydrogen bonds, with an average of 5 hydrogen bonds, indicating its favorable binding affinity compared to other complexes studied.Gibb's free energy evaluation suggested the existence of more favorable low-energy conformations of AMP1 and AMP12b bound histone demethylase.

For sortin, the complex with AMP12a displayed the most favorable low-energy conformations, indicating its stability. Similarly, squalene synthase bound to AMP4 exhibited the lowest energy conformations, suggesting potentially favorable affinity. However, DSSP analysis revealed minimal secondary structural changes in proteins such as histone demethylase, sortin, and squalene synthase throughout the simulation. Additionally, minor secondary changes were observed in the structure of individual AMPs throughout the simulation period. The DCC analysis is crucial in identifying the key positively correlating residues. It pointed out that the AMP1, AMP3, and AMP12b interacted with histone demethylase residues. Similarly, AMP12a and AMP15 made key interactions with sortin residues. In the case of squalene synthase, AMP3, AMP4, and AMP12b made key interactions. All these complexes may be more favourable than the remaining complexes for each protein. On the other hand, the MM-PBSA analysis revealed that AMP10 had the lowest binding energy for histone demethylase and sortin, while AMP6 had the lowest binding energy against squalene synthase. For these peptides, the lowest electrostatic interaction energy may be responsible for the lowest binding energy.

## Conclusion

Our research highlights the potential of antifungal peptide extracts sourced from *S. tuberosum* and *C. annum* in combating *R. solani*. Specifically, our results reveal that the peptide extract from *C. annum*, at a concentration of 50 mg/mL, demonstrates both fungistatic and fungicidal effects against *R. solani*, while a concentration of 25 mg/mL of *S. tuberosum* extract proves effective for both purposes. However, our docking analysis unveils that not all peptides form suitable complexes with the modeled pathogenicity-related proteins from *R. solani*, as indicated by unfavorable binding free energy. Notably, AMP10 exhibits favorable binding energy for all three PR-proteins, while AMP6 demonstrates the highest potential against squalene synthase. Moving forward, further research is imperative, particularly focusing on both *in vivo* and *in vitro* applications of the identified peptides. Such investigations hold promise for the development of novel and environmentally friendly peptide antifungal agents derived from plants. These natural potential antifungal agents have the capacity to significantly contribute to the control and management of *R. solani*, thereby alleviating the adverse impact on rice production.

## Acknowledgments

The support provided by the International Foundation for Collaborative Research (IFCR) and LifeQuest Research Academy (LRA), is duly recognized by the authors.

## Author Contributions

**Conceptualization:** Tijjani Mustapha, Bhoomendra A. Bhongade.

**Formal analysis:** Tijjani Mustapha, Shefin B.

**Investigation:** Tijjani Mustapha, Shefin B, Rajesh B. Patil, Bhoomendra A. Bhongade.

**Methodology:** Tijjani Mustapha, Shefin B, Rajesh B. Patil, Bhoomendra A. Bhongade, Jaiprakash N. Sangshetti, Aniket Mali.

**Project administration:** Bhoomendra A. Bhongade, Abu Tayab Moin.

**Software:** Rajesh B. Patil.

**Supervision:** Tofazzal Islam.

**Visualization:** Rajesh B. Patil.

**Writing – original draft:** Tijjani Mustapha, Shefin B, Rajesh B. Patil, Bhoomendra A. Bhongade, Jaiprakash N. Sangshetti, Aniket Mali.

**Writing – review & editing:** Talha Zubair, Balogun Joshua Babalola, Abu Tayab Moin, Tofazzal Islam.

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
