## [Decision Letter · Decision Letter 0]

4 Oct 2023

PONE-D-23-28472Antimicrobial Peptides from Solanaceae Plants: Analyzing Their Effect on Phytopathogenic Fungus Rhizoctonia solaniPLOS ONE

Dear Dr. Islam,

Thank you for submitting your manuscript to PLOS ONE. After careful consideration, we feel that it has merit but does not fully meet PLOS ONE’s publication criteria as it currently stands. Therefore, we invite you to submit a revised version of the manuscript that addresses the points raised during the review process.

Please review the comments provided by the reviewers and address their feedback. Additionally, we recommend seeking a professional English editor to ensure the manuscript is free of language errors. Once these steps are taken, kindly submit your revised manuscript.

We look forward to receiving your revised manuscript.

Kind regards,

Kamal Ahmad Qureshi, PhD

Academic Editor

PLOS ONE

Journal Requirements:

Reviewers' comments:

Reviewer's Responses to Questions

**Comments to the Author**

1. Is the manuscript technically sound, and do the data support the conclusions?

Reviewer #1: Yes

Reviewer #2: Yes

2. Has the statistical analysis been performed appropriately and rigorously? 

Reviewer #1: Yes

Reviewer #2: Yes

3. Have the authors made all data underlying the findings in their manuscript fully available?

Reviewer #1: Yes

Reviewer #2: Yes

4. Is the manuscript presented in an intelligible fashion and written in standard English?

Reviewer #1: Yes

Reviewer #2: No

5. Review Comments to the Author

Reviewer #1: The authors of a study have suggested that peptides derived from Solanaceous plants like Solanum tuberosum and Capsicum annum have antifungal properties against R. solani, a fungal pathogen. They have also performed molecular docking studies of these peptides with pathogenicity-related proteins of R. solani. The study did not present any new scientific hypothesis or concept. Therefore, my comments are:

There are some suggestions for improvement. Firstly, the abstract needs to be shortened to only present the silent findings of the research outcome. Secondly, performing q-PCR of blight resistance and sensitivity-related genes from treated and non-treated samples with peptides would validate the findings.

In the material method section, it is suggested to provide the full genus name as the first time appearing in the manuscript for C. annum and S. tuberosum. Additionally, it is unclear if the author quantified the level of peptide and identified any. The extraction buffer used for peptide extraction should be mentioned too.

Barashkova and Rogozhin, 2020 is a review article, so it should be clarified if the author made any modifications or standardized the protocol. Moreover, it is unclear how the author deduced that the extract contains peptides that were used for docking.

Lastly, the author needs to justify why they have selected prp Histone demethylase. The paper needs to be reorganized and proofread before submission.

Reviewer #2: The present article entitled "Antimicrobial Peptides from Solanaceae Plants: Analyzing Their Effect on

Phytopathogenic Fungus Rhizoctonia solani" based on a good theme and authors have briefly discussed all the point. The presenation of the article is also fine.

How ever I have only suggestion to improve the English of the article

6. PLOS authors have the option to publish the peer review history of their article (what does this mean?). If published, this will include your full peer review and any attached files.

Reviewer #1: **Yes: **Prashant Kumar Singh

Reviewer #2: No

---

## [Author Response · Author response to Decision Letter 0]

12 Dec 2023

Responses to the reviewers' comments

Reviewer #1:

Reviwer’s comment: The authors of a study have suggested that peptides derived from Solanaceous plants like Solanum tuberosum and Capsicum annum have antifungal properties against R. solani, a fungal pathogen. They have also performed molecular docking studies of these peptides with pathogenicity-related proteins of R. solani. The study did not present any new scientific hypothesis or concept. Therefore, my comments are:

There are some suggestions for improvement. Firstly, the abstract needs to be shortened to only present the silent findings of the research outcome. 

Authors’ response: Thank you for the critical comment and valuable suggestion for the improvement of the manuscript. We substantially revised the manuscript and focused the novelty and perspective of our findings. The abstract is also shortened as suggested.

Reviwer’s comment: Secondly, performing q-PCR of blight resistance and sensitivity-related genes from treated and non-treated samples with peptides would validate the findings.

Authors’ response: We agree with the reviewer that q-PCR of blight resistance and sensitivity related genes from treated and untreated samples with peptides would validate our in vitro and insilico research findings. However, this approach was not the scope of the current study and it takes long time and resources. We are now planning to include this important idea in our new project as a follow up this study.

Reviwer’s comment: In the material method section, it is suggested to provide the full genus name for the first time appearing in the manuscript for C. annum and S. tuberosum. 

Authors’ response: The full genus names are included for the first time appearing of C. annum and S. tuberosum, in the revised manuscript. Thank you so much for the comment. 

Reviwer’s comment: Additionally, it is unclear if the author quantified the level of peptide and identified any. 

Authors’ response: Thank you very much for your comment. Regrettably, our current laboratory capabilities and expertise in isolating peptides from plant extracts are very limited. Consequently, we have not undertaken quantification or characterization of antimicrobial peptide levels in peptide-rich fractions of C. annum and S. tuberosum leaves. We acknowledge the importance of this aspect and hope to explore collaborations or acquire necessary resources to address this in future studies. Your understanding of these constraints is greatly appreciated.

Reviwer’s comment: The extraction buffer used for peptide extraction should be mentioned too.

Authors’ response: Thank you very much for your comment. The optimized method reviewed by Barashkova & Rogozhin, 2020 and the method described by Astafieva et al., 2012 and Taveira et al., 2014 were adopted. This method does not require the use of a buffer. 

Reviwer’s comment: Barashkova and Rogozhin, 2020 is a review article, so it should be clarified if the author made any modifications or standardized the protocol. 

Authors’ response: Thank you very much for your valuable comment. We have adopted the protocol from the cited literature in this review (Astafieva et al., 2012; Taveira et al., 2014). Further, these references are cited in the methods section of the revised manuscript. 

Reviwer’s comment: Moreover, it is unclear how the author deduced that the extract contains peptides that were used for docking.

Authors’ response: Thank you very much for your valuable comment. We deduced the presence of the peptides in peptide-rich extracts from C. annum and S. tuberosum leaves from the literature Afroz et al., 2020; Dev & Venu, 2016; Maracahipes et al., 2019; Moulin et al., 2014. 

Reviwer’s comment: Lastly, the author needs to justify why they have selected prp Histone demethylase. 

Authors’ response: Thank you very much for your valuable comment. Histone demethylase has been suggested as a target protein affecting biological processes such as pathogenicity, virulence, vegetative growth, and morphogenesis in R. solani (Prabhukarthikeyan et al., 2022). This compelled us to undertake a study on Histone demethylase. 

Reviwer’s comment: The paper needs to be reorganized and proofread before submission.

Authors’ response: We agree with the valuable comment of the reviewer and hence substantially revised and reorganized the manuscript for coherence and clarity of the novelty and perspective of the findings. The language of the manuscript is checked thoroughly.

Reviewer #2:

Reviwer’s comment: The present article entitled "Antimicrobial Peptides from Solanaceae Plants: Analyzing Their Effect on Phytopathogenic Fungus Rhizoctonia solani" based on a good theme and authors have briefly discussed all the point. The presenation of the article is also fine. 

Authors’ response: We sincerely appreciate the reviewer for this encouraging comment on our manuscript.

Reviwer’s comment: How ever I have only suggestion to improve the English of the article

Authors’ response: The manuscript was thoroughly proofread and substantially revised, including improving the English usage. Thank you very much for your valuable comment.

---

## [Decision Letter · Decision Letter 1]

11 Mar 2024

PONE-D-23-28472R1Antimicrobial Peptides from Solanaceae Plants: Analyzing Their Effect on Phytopathogenic Fungus Rhizoctonia solaniPLOS ONE

Dear Dr. Islam,

Thank you for submitting your manuscript to PLOS ONE. After careful consideration, we feel that it has merit but does not fully meet PLOS ONE’s publication criteria as it currently stands. Therefore, we invite you to submit a revised version of the manuscript that addresses the points raised during the review process.

We look forward to receiving your revised manuscript.

Kind regards,

Kamal Ahmad Qureshi, PhD

Academic Editor

PLOS ONE

Journal Requirements:

**Additional Editor Comments:**

**Note: **Kindly address the comments of our respected reviewers. 

Reviewers' comments:

Reviewer's Responses to Questions

**Comments to the Author**

1. If the authors have adequately addressed your comments raised in a previous round of review and you feel that this manuscript is now acceptable for publication, you may indicate that here to bypass the “Comments to the Author” section, enter your conflict of interest statement in the “Confidential to Editor” section, and submit your "Accept" recommendation.

Reviewer #1: All comments have been addressed

Reviewer #3: (No Response)

Reviewer #4: All comments have been addressed

Reviewer #5: All comments have been addressed

2. Is the manuscript technically sound, and do the data support the conclusions?

Reviewer #1: Yes

Reviewer #3: (No Response)

Reviewer #4: No

Reviewer #5: Partly

3. Has the statistical analysis been performed appropriately and rigorously? 

Reviewer #1: Yes

Reviewer #3: (No Response)

Reviewer #4: Yes

Reviewer #5: N/A

4. Have the authors made all data underlying the findings in their manuscript fully available?

Reviewer #1: Yes

Reviewer #3: (No Response)

Reviewer #4: Yes

Reviewer #5: Yes

5. Is the manuscript presented in an intelligible fashion and written in standard English?

Reviewer #1: Yes

Reviewer #3: (No Response)

Reviewer #4: No

Reviewer #5: Yes

6. Review Comments to the Author

Reviewer #1: Authors have resolved the all my concern related to MS. Therefore, my decision to accept the ms for publication.

Reviewer #3: The manuscript is of wrathful consideration for plant scientists. This is based upon effective study Antimicrobial Peptides from Solanaceae Plants: Analyzing, Their Effect on Phytopathogenic Fungus Rhizoctonia solan in that region. The write-up of whole manuscript is up to mark. English quality is acceptable but minor revision needed. However, there are some suggestions that need to be incorporated before final acceptance as elaborated below:

- Title of the manuscript should be changed.

- The Abstract needs few details about materials of the experiments.

- Please note the opening paragraph of the introduction could provide stronger context to the paper, and, similarly, the findings at the end could potentially be richer.

- At the end of Introduction section, there should be clear hypothesis and objectives of the designed study.

- Statistical designs and data analysis confusing. It needs to present in a proper way

- Results presentations should be revised profoundly.

- I suggest that the author should provide more justification for your study (specifically, you should expand upon the knowledge gap in the abstract, introduction, and all other sections being filled) which should be improved upon before Acceptance.

- The discussion need to revised and need to make it more focused based on results.

Reviewer #4: (No Response)

Reviewer #5: Reviewer’s comment

The manuscript is written on potential antimicrobial peptides for the control of Rhizoctonia solani, causing rice sheath blight. Corrections are suggested below.

In abstract: A statement need to be included the confirmation of presence of peptides in the extracts.

In materials and method, et al., is written in italics. Should be same throughout the manuscript. The field details (name, district, latitude, longitude etc.) under source of inoculum should be mentioned. Space should be given between 5 mm. 2.3 and 2.4 should be merged. Hour should be written as h. There should be space between ° and C.

How do you know the extract is peptide rich? Include suitable references indicating peptide extraction protocol. Otherwise, characterization needs to be done.

In-vitro should be in italics.

2.11 Once Rhizoctonia solani written in abstract, no need to mention complete name, only write R. solani should be written throughout the manuscript.

What are the names of the peptides present in the peptide rich extract? How do you prepare their structures? Mention the names and structures if someone has already been identified them previously from the plants taken.

2.12. Molecular dynamics studies and MM-PBSA calculations section is very elaborative. It should be written concisely. Use references and avoid detail process.

For statistical analysis, atleast five test concentrations should be taken along with positive and negative control with three or more replications.

What was the positive control. Please mention the name and data in the table 1 and 2.

Table 4. Mention the source (reference from where the sequences of the peptides retrieved). The names of enzymes in Table 4 should be properly written. Write the unit of binding energy (kcal).

7. PLOS authors have the option to publish the peer review history of their article (what does this mean?). If published, this will include your full peer review and any attached files.

Reviewer #1: **Yes: **PRASHANT KUMAR Singh

Reviewer #3: No

Reviewer #4: No

Reviewer #5: No

---

## [Author Response · Author response to Decision Letter 1]

23 Mar 2024

Responses to the reviewers’ comments

Reviewers' comments:

Reviewer #1

Reviewer #1: Authors have resolved the all my concern related to MS. Therefore, my decision to accept the ms for publication.

Response: Thank you for this encouraging decision.

Reviewer #3

Reviewer #3: The manuscript is of wrathful consideration for plant scientists. This is based upon effective study Antimicrobial Peptides from Solanaceae Plants: Analyzing, Their Effect on Phytopathogenic Fungus Rhizoctonia solan in that region. The write-up of whole manuscript is up to mark. English quality is acceptable but minor revision needed. 

Response: Thank you. We revised the English usage, making the manuscript worthy. 

However, there are some suggestions that need to be incorporated before final acceptance as elaborated below:

- Title of the manuscript should be changed.

Response: The title is revised as “In-vitro and In-silico Investigation of Effects of Antimicrobial Peptides from Solanaceae Plants on Rice Sheath Blight Pathogen”.

- The Abstract needs few details about materials of the experiments.

Response: This valid observation has been addressed in the abstract section of the MS as suggested.

- Please note the opening paragraph of the introduction could provide stronger context to the paper, and, similarly, the findings at the end could potentially be richer.

Response: The opening paragraph has already captured the context of the paper as it states the damaging effect of R. solani and possible yield loss due to the sheath blight pathogen infection. This explains the focus of the research for unveiling a safe method of the R. solani control.

- At the end of Introduction section, there should be clear hypothesis and objectives of the designed study.

Response: Thank you. The clear hypothesis has been incorporated at the end of the introduction section.

- Statistical designs and data analysis confusing. It needs to present in a proper way

Response: Statistical design has been revised and explained in a separate section 2.13 of the MS.

- Results presentations should be revised profoundly.

Response: We appreciate you for this encouraging comment. But the results were expressed in the best way it should be presented. This is evident when the figures and tables are observed.

- I suggest that the author should provide more justification for your study (specifically, you should expand upon the knowledge gap in the abstract, introduction, and all other sections being filled) which should be improved upon before Acceptance.

Response: the justification of the study, particularly in the introduction section, has been provided.

- The discussion need to revised and need to make it more focused based on results.

Response: Thank you. The discussion was rigorously based on the focus and results obtained from the study. These have been addressed from the beginning of the MS preparation.

Reviewer #4

Reviewer #4: (No Response)

Reviewer #5

Reviewer #5: Reviewer’s comment

The manuscript is written on potential antimicrobial peptides for the control of Rhizoctonia solani, causing rice sheath blight. Corrections are suggested below.

In abstract: A statement need to be included the confirmation of presence of peptides in the extracts.

Response: Thank you. An optimized method of AMP extraction was used as an extraction procedure in this study, according to these authors (Astafieva et al., 2012; Barashkova & Rogozhin, 2020; Taveira et al., 2014). Therefore, this requires no further confirmation of the AMPs extracted due to the notion that the extract obtained using this method contains AMPs. The plants used as a source of the AMPs were confirmed to possess these peptides, as established by Afroz et al. (2020). 

In materials and method, et al., is written in italics. Should be same throughout the manuscript.

Response: All ‘et al’ in the updated MS have been italicized as suggested.

The field details (name, district, latitude, longitude etc.) under source of inoculum should be mentioned. Space should be given between 5 mm. 2.3 and 2.4 should be merged. Hour should be written as h. There should be space between ° and C.

Response: Field details have been provided in the MS. 2.3 and 2.4 were merged. All issues raised here have been addressed.

How do you know the extract is peptide rich? Include suitable references indicating peptide extraction protocol. Otherwise, characterization needs to be done.

Response: The plants used as a source of the AMPs were confirmed to possess these peptides as established by Afroz et al. (2020). 

In-vitro should be in italics.

Response: Correction is made in the revised manuscript.

2.11 Once Rhizoctonia solani written in abstract, no need to mention complete name, only write R. solani should be written throughout the manuscript.

Response: The correction is done in the revised manuscript as suggested. 

What are the names of the peptides present in the peptide rich extract? How do you prepare their structures? Mention the names and structures if someone has already been identified them previously from the plants taken.

Response: The structures of AMP from the Solanaceae plants were predicted from the PEP-FOLD3 server (Lamiable et al., 2016; Shen et al., 2014). 

2.12. Molecular dynamics studies and MM-PBSA calculations section is very elaborative. It should be written concisely. Use references and avoid detail process.

Response: We agree with the concerns the reviewer raised. Wherever needed, the content is made concise. However, we feel that the details, whichever are kept in the revised manuscript, will give a deep understanding of the methodology and enhance the repeatability of the method.

For statistical analysis, atleast five test concentrations should be taken along with positive and negative control with three or more replications.

Response: This was what exactly was done during the analysis phase of the data obtained.

What was the positive control. Please mention the name and data in the table 1 and 2.

Response: DMSO was used as a negative control throughout the bioassay, while Azoxystrobin, a broad-spectrum systemic fungicide, was used as a positive control. Tables 1 and 2 were inserted in the positive control.

Table 4. Mention the source (reference from where the sequences of the peptides retrieved).

Response: This was fully mentioned in the “Protein-Peptide Docking Analysis” section of the manuscript.

The names of enzymes in Table 4 should be properly written. Write the unit of binding energy (kcal).

Response: We appreciate you for the kind observation. These details have been provided in the Table in the revised manuscript.

---

## [Decision Letter · Decision Letter 2]

4 Apr 2024

In-vitro and In-silico Investigation of Effects of Antimicrobial Peptides from Solanaceae Plants on Rice Sheath Blight Pathogen

PONE-D-23-28472R2

Dear Dr. %Tofazzal Islam%,

We’re pleased to inform you that your manuscript has been judged scientifically suitable for publication and will be formally accepted for publication once it meets all outstanding technical requirements.

Kind regards,

Kamal Ahmad Qureshi, PhD

Academic Editor

PLOS ONE

Additional Editor Comments (optional):

Reviewers' comments:

Reviewer's Responses to Questions

**Comments to the Author**

1. If the authors have adequately addressed your comments raised in a previous round of review and you feel that this manuscript is now acceptable for publication, you may indicate that here to bypass the “Comments to the Author” section, enter your conflict of interest statement in the “Confidential to Editor” section, and submit your "Accept" recommendation.

Reviewer #1: All comments have been addressed

Reviewer #5: All comments have been addressed

2. Is the manuscript technically sound, and do the data support the conclusions?

Reviewer #1: Yes

Reviewer #5: Yes

3. Has the statistical analysis been performed appropriately and rigorously? 

Reviewer #1: I Don't Know

Reviewer #5: N/A

4. Have the authors made all data underlying the findings in their manuscript fully available?

Reviewer #1: Yes

Reviewer #5: (No Response)

5. Is the manuscript presented in an intelligible fashion and written in standard English?

Reviewer #1: Yes

Reviewer #5: Yes

6. Review Comments to the Author

Reviewer #1: Authors have already complied my comments and concerns raised. Hence MS is in shape and can be accepted to plos one

Reviewer #5: Since the suggested correction have been incorporated in the revised manuscript thus, the manuscript may be accepted for publication.

7. PLOS authors have the option to publish the peer review history of their article (what does this mean?). If published, this will include your full peer review and any attached files.

Reviewer #1: **Yes: **Prashant Kumar Singh

Reviewer #5: No

---

## [Editor Report · Acceptance letter]

3 Jun 2024

PONE-D-23-28472R2 

PLOS ONE

Dear Dr. Islam, 

I'm pleased to inform you that your manuscript has been deemed suitable for publication in PLOS ONE. Congratulations! Your manuscript is now being handed over to our production team.

Kind regards, 

on behalf of

Dr. Kamal Ahmad Qureshi 

Academic Editor

PLOS ONE